# Can Mamba Always Enjoy the "Free Lunch"?

## Abstract

Transformers have been the cornerstone of current Large Language Models (LLMs); however, its linear growth in overhead during inference with respect to sequence length poses challenges for modeling long sequences. In this context, Mamba has gradually attracted attention due to its constant-level size during inference and existing empirical results have shown that it can perform comparably to Transformers in sequence modeling while offering significant savings. However, one may ask that, can Mamba always enjoy the "free lunch"? In this paper, we focus on analyzing the expressive ability of Mamba from a theoretical standpoint. First, inspired by the connection between Mamba and linear attention, we investigate potential shortcomings of the Mamba when performing the COPY operation. Our results indicate that Mamba with constant size may encounter bottlenecks when handling COPY, while it can achieve perfect performance when the size scales linearly with sequence length. Based on this observation, we analyze Mamba's ability to tackle DP problems when equipped with Chain of Thought (CoT). Our findings suggest that to solve arbitrary DP problems, the total cost of Mamba is comparable to standard and efficient Transformers. However, similar to efficient Transformers, when facing DP problems with favorable properties such as locality, Mamba can provide savings in overhead. Our results contribute to a deeper understanding of Mamba.

## 1 Introduction

Reccently, Transformer-based large language models (LLMs) have become the mainstream of modern neural network architectures due to their outstanding performance across a wide range of tasks (Vaswani et al., 2017; Kenton & Toutanova, 2019; Brown et al., 2020; Dosovitskiy et al., 2020; Min et al., 2022). However, the core component of Transformers—the attention layer—while providing excellent performance, also leads to emerging drawbacks: during training, the computational cost scales quadratically with sequence length, and during inference, the cost scales linearly with sequence length. This limitation becomes increasingly unacceptable when dealing with long sequence tasks. To address this issue, many works have attempted to improve the attention mechanism to reduce its time and storage costs (Tay et al., 2023; Choromanski et al., 2020; Katharopoulos et al., 2020; Beltagy et al., 2020; Child et al., 2019). However, these improved structures often achieve efficiency in the attention layer at the expense of some performance.

Faced with the scaling challenges of Transformers, the exploration of new model architectures to replace Transformers has gradually come into focus, leading to the development of modern RNN architectures, including RWKV (Peng et al., 2023), RetNet (Sun et al., 2023), and Mamba (Gu & Dao, 2023). Among them, the Mamba architecture (Gu et al., 2021; Gu & Dao, 2023), based on the state space model (SSM), has garnered attention for its performance comparable to Transformers in many sequence modeling tasks Dao & Gu (2024) and vision tasks (Zhu et al., 2024; Xu et al., 2024). These models utilize hardware-aware algorithms during training, resulting in computational costs that scale linearly with sequence length, and require constant-level computation and storage during inference. Mamba's strong performance and computational efficiency make it a strong competitor to Transformers.

Despite Mamba demonstrating excellent performance, one can not help but ask, *can Mamba always enjoy such "free lunch", that is, can Mamba always save considerable overhead while maintaining performance of Transformers?* More recent results have revealed Mamba's shortcomings in certain tasks, especially those involving retrieval (Arora et al., 2023; Hendrycks et al., 2020; Jelassi et al.,

2024). Specifically, Akyürek et al. (2024) study the in-context language learning capabilities of different models and find that Transformers outperformed other models, including Mamba, due to the specialized attention heads. Jelassi et al. (2024) also discover that Transformers are superior to Mamba on tasks that require copying from the input context. Park et al. (2024) point out that Mamba struggles to retrieve vectors from the context of multi-query associative recall (MQAR) (Arora et al., 2023), while Transformers can easily handle it well. Furthermore, Waleffe et al. (2024) conduct experiments on larger models (up to 8B parameters) with a broader range of tasks, discovering that when it comes to in-context learning and recalling information from text, although Mambas can contain the same knowledge as Transformers, it will be more difficult for them to directly copy useful information from history.

Although there has been some empirical exploration, the theoretical investigation concerning the above "free lunch" question still remains open to explore. In this paper, we attempt to explore Mamba's expressive ability from a theoretical standpoint. Specifically, inspired by the comparison between Mamba and linear attention mechanism, we first focus on Mamba's ability to perform the COPY operation, which is closely related to the ability to retrieve information from context. Our theoretical results show that Mamba's performance in executing COPY operations is closely related to the size of its model and experiments also confirm the limitations of Mamba in executing copy tasks and retrieval from long sequences. Further, following the setting of Feng et al. (2024); Yang et al. (2024), we explore Mamba's capability to solve DP tasks equipped with Chain of Thought (CoT). We find that to solve arbitrary DP problems, Mamba, standard Transformers and efficient Transformers including Linear and Sparse Transformers (Katharopoulos et al., 2020; Child et al., 2019) seem to be on equal footing in terms of inference cost; however, when dealing with $m$-locality DP problems, Mamba may offer significant savings like efficient Transformers. Our results can be concluded as follows:

- A constant-sized Mamba may encounter bottlenecks when executing COPY operations (Theorem 1 in Section 4);

- When the size of Mamba scales linearly with the sequence length, it can perform COPY operation accurately (Theorem 2 in Section 4);

- To solve arbitrary DP problems, the total cost required by Mamba is comparable to that of standard and efficient Transformers (Theorem 3 in Section 5);

- When dealing with DP problems that have favorable locality properties, Mamba can bring savings in overhead compared to standard Transformers (Theorem 4 in Section 5).

## 2 RELATED WORK

**SSMs and Attention Mechanism:** The attention mechanism is a core component of LLMs (Brown et al., 2020; Touvron et al., 2023). Drawing connections between SSMs and attention is a fascinating direction as it not only aids in our understanding of the Mamba structure but also facilitates the transfer of well-established acceleration techniques from attention mechanisms to Mamba (Dao, 2023; Katharopoulos et al., 2020). Based on observations of the similarities between them, Dao & Gu (2024) proposed the state space dual (SSD) layer based on SSMs to achieve significant improvements in training efficiency. Sieber et al. (2024) introduce the Dynamical Systems Framework (DSF), under which attention and SSMs can be directly compared. Additionally, Han et al. (2024) reformulate the structure of SSMs to establish links with linear attention, aiming to investigate the key factors behind success in vision tasks. We follow this convenient reformulation and comparison, based on which we furthermore explore Mamba's ability to perform the COPY operation.

**Comparisons between Transformers and Mamba:** More recent works compare the performance of Mamba and Transformers across various tasks from different perspectives. Merrill et al. (2024) theoretically demonstrate that, similar to Transformers, Mamba is also unable to solve state tracking problems such as permutation composition. Jelassi et al. (2024) find that Transformers significantly surpass SSMs when facing tasks related to copying and retrieving information from context. Park et al. (2024) investigate Mamba's capability for in-context learning and demonstrate that Mamba outperforms Transformers in sparse parity learning while it is weaker in tasks involving non-standard retrieval functionality. Similarly, Waleffe et al. (2024) conduct experiments on a larger scale and find that Mamba lag behind Transformers in tasks that require strong copying and long-context

reasoning. Our experiments reference the setups of these works and conduct similar investigations. We also note that Jelassi et al. (2024) find theoretical conclusions that appear to be similar to ours, namely that generalized SSMs (GSSMs) including Mamba cannot copy uniformly input sequences unless the size of the state space grows linearly with the sequence length. Although our work also focuses on "copy", Jelassi et al. (2024)'s definition of the copy task is at the sequence level, investigating the ability of GSSMs to replicate entire sequences that satisfy some distribution and providing a lower bound for their state space memory. In contrast, our COPY operation is defined at the token level, exploring the conditions under which Mamba can copy some specific historical token and providing constructions for Mamba with linear-scaling size to achieving this operation.

**Transformers and modern RNNs with CoT:** Chain-of-Thought (CoT) (Wei et al., 2022) is employed to enhance the performance of LLMs by enabling them to provide step-by-step reasoning before arriving at a final answer. It has been shown theoretically that Transformers with CoT exhibit significantly improved expressive power, allowing them to solve more complex problems compared to Transformers without CoT (Merrill & Sabharwal, 2023b; Feng et al., 2024; Merrill & Sabharwal, 2023a; Li et al., 2024; Yang et al., 2024). Our analysis of Mamba equipped with CoT follows the framework set by Feng et al. (2024); Yang et al. (2024) in their analysis of dynamic programming (DP) problems. Furthermore, Wen et al. (2024) examine the effect of CoT on enhancing the expressive power of modern RNNs including SSMs by drawing connections with a Turing machine with $O(log(n))$ space. Their findings indicate that when using CoT, RNNs with $log(n)$ bit memory will have strictly stronger representation power than those without CoT. Different from this, our work explores the ability of Mamba equipped with CoT from the perspective of solving DP problems following the setting of Feng et al. (2024) and show constructions for Mamba layers with linear-scaling size relative to the sequence length to solve DP problems.

## 3 PRELIMINARIES

In this section, we introduce the Mamba structure that we focus on and its reformulated form firstly introduced by Han et al. (2024), which facilitates a better understanding of the connection between Mamba and linear attention as illustrated in Section 4.1. It should be noted that to better distinguish different types of variables, in this paper, we use bold uppercase letters to represent matrices such as $\boldsymbol{A}$, bold lowercase letters to represent vectors such as $\boldsymbol{a}$, and all non-bold letters to represent scalars such as $a$ and $\Delta$. This may differ slightly from the notations used in some Mamba-related literatures (Dao & Gu, 2024; Zhu et al., 2024), where uppercase letters are used to describe $\boldsymbol{A}, \boldsymbol{B}, \boldsymbol{C}$ in SSMs.

**State Space Model:** The state space model (SSM) is inspired by the continuous system that maps a scalar input $x(t) \in \mathbb{R}$ to its output $y(t) \in \mathbb{R}$ through a high-dimensional hidden state $\boldsymbol{h} \in \mathbb{R}^{d_h}$ (Gu & Dao, 2023; Dao & Gu, 2024; Han et al., 2024; Zhu et al., 2024; Han et al., 2024). Specifically, this system can be written as:

$$\boldsymbol{h}'(t) = \boldsymbol{A}\boldsymbol{h}(t) + \boldsymbol{b}x(t), \quad y(t) = \boldsymbol{c}^T\boldsymbol{h}(t) + dx(t),$$

where $\boldsymbol{A} \in \mathbb{R}^{d_h \times d_h}$ denotes the evolution parameters, $\boldsymbol{b}, \boldsymbol{c} \in \mathbb{R}^{d_h}$ are projection parameters and $d$ is a scalar parameter. The above continuous system can be discretized using transformation called zero-order hold (ZOH), resulting in a discrete version that can be used for neural networks. In this process, $\boldsymbol{A}, \boldsymbol{b}$ will be transformed as $\overline{\boldsymbol{A}}, \overline{\boldsymbol{b}}$. The discrete version for SSM can be written as:

$$\boldsymbol{h}_i = \overline{\boldsymbol{A}}\boldsymbol{h}_{i-1} + \overline{\boldsymbol{b}}x_i, \quad y_i = \boldsymbol{c}^T\boldsymbol{h}_i + dx_i,$$

where $\overline{\boldsymbol{A}} = \exp(\Delta\boldsymbol{A})$, $\overline{\boldsymbol{b}} = (\Delta\boldsymbol{A})^{-1}(\exp(\Delta\boldsymbol{A}) - \boldsymbol{I}) \cdot \Delta\boldsymbol{b} \approx \Delta\boldsymbol{b}$ and $\Delta \in \mathbb{R}$ is a timescale parameter. The matrix $\boldsymbol{A}$ is typically assumed to have certain structures such as being diagonal, leading to the structured SSMs (Gu et al., 2022; Gupta et al., 2022).

**Selective State Space Module:** To enhance the SSM, Mamba makes the parameters $\boldsymbol{b}_i, \boldsymbol{c}_i, \Delta_i$ dependent on different inputs $x_i$. More specifically, $\boldsymbol{A}$ is set to be diagonal resulting in that $\overline{\boldsymbol{A}}_i\boldsymbol{h}_{i-1} = \tilde{\boldsymbol{a}}_i \odot \boldsymbol{h}_{i-1}$ where $\tilde{\boldsymbol{a}}_i = \exp(\Delta_i\boldsymbol{a})$, $\boldsymbol{a} = \mathrm{diag}(\boldsymbol{A})$ and $\odot$ denotes the element-wise product. In addition, $\overline{\boldsymbol{b}}_ix_i = \Delta_i\boldsymbol{b}_ix_i = \boldsymbol{b}_i(\Delta_i \odot x_i)$. Thus, this transformation ultimately results in:

$$\boldsymbol{h}_i = \tilde{\boldsymbol{a}}_i \odot \boldsymbol{h}_{i-1} + \boldsymbol{b}_i(\Delta_i \odot x_i), \quad y_i = \boldsymbol{c}_i^T\boldsymbol{h}_i + d \odot x_i.$$

Furthermore, to extend the case of processing scalar inputs $x_i$ to vectors $\boldsymbol{x}_i \in \mathbb{R}^d$, Mamba performs the above operations on each dimension independently, which can be formalized as:

$$\boldsymbol{H}_i = \widetilde{\boldsymbol{A}}_i \odot \boldsymbol{H}_{i-1} + \boldsymbol{b}_i(\boldsymbol{\Delta}_i \odot \boldsymbol{x}_i)^T, \quad \boldsymbol{y}_i = \boldsymbol{H}_i^T\boldsymbol{c}_i + \boldsymbol{d} \odot \boldsymbol{x}_i, \tag{1}$$

where $\widetilde{\boldsymbol{A}}_i = [\tilde{\boldsymbol{a}}_i^{(j)}]_{j=1}^d \in \mathbb{R}^{d_h \times d}$, $\boldsymbol{b}_i = \boldsymbol{W_b}\boldsymbol{x}_i \in \mathbb{R}^{d_h}$, $\boldsymbol{c}_i = \boldsymbol{W_c}\boldsymbol{x}_i \in \mathbb{R}^{d_h}$, $\boldsymbol{\Delta}_i = \text{Softplus}(\boldsymbol{W_\Delta^2}\boldsymbol{W_\Delta^1}\boldsymbol{x}_i) \in \mathbb{R}^d$ and $\boldsymbol{W_\Delta^1}, \boldsymbol{W_\Delta^2}$ are linear projection parameters. Thus, given the input $\boldsymbol{X} = [\boldsymbol{x}_i]_{i=1}^N \in \mathbb{R}^{d \times N}$, we denote the output of the SSM module in Mamba as $\boldsymbol{Y} = \text{SSM}(\boldsymbol{X})$ where $\boldsymbol{Y} = [\boldsymbol{y}_i]_{i=1}^N$ and $\boldsymbol{y}_i$ follows Eq (1). This formalization was introduced by Han et al. (2024) to build a bridge between Mamba and linear attention and here we follow this form.

**Mamba Layer:** Given some input sequence $\boldsymbol{X} = [\boldsymbol{x}_i]_{i=1}^N \in \mathbb{R}^{d \times N}$, the input will be processed through stacked Mamba layers where each layer can be viewed as consisting of a residual connection and a Mamba block $\boldsymbol{f}^{(l)} : \mathbb{R}^d \to \mathbb{R}^d$. Specifically, this process can be formulated as

$$\boldsymbol{X}_l = \boldsymbol{X}_{l-1} + \boldsymbol{f}^{(l)}(\boldsymbol{X}_{l-1}), \quad l = 1, 2, \ldots, L \tag{2}$$

$$\boldsymbol{f}^{(l)}(\boldsymbol{X}_{l-1}) = \boldsymbol{W}_3^{(l)} \cdot \text{SSM}(\boldsymbol{W}_1^{(l)}\boldsymbol{X}_{l-1} + \boldsymbol{b}_1^{(l)}) \odot \sigma(\boldsymbol{W}_2^{(l)}\boldsymbol{X}_{l-1} + \boldsymbol{b}_2^{(l)}), \tag{3}$$

where $\sigma(\cdot)$ denotes SiLU activation function. A Mamba block combines the output of the SSM module with a gated MLP (Gu & Dao, 2023; Chowdhery et al., 2023; Shazeer, 2020), which can be viewed as consisting of two branches. Here we call the branch with the SSM module as "the SSM branch" and the other as "the gated branch". It should be noticed that for the sake of simplifying the analysis, we focus on the core components of Mamba including SSM module and the gated MLP while ignoring other structures including the $\sigma(\cdot)$ before SSM module, Layer Normalization and 1D Convolutions. The simplified Mamba structure that we focus on can be illustrated by Figure 1. In addition, to avoid confusion, we clarify that the SSM module here is specifically the Selective State Space Module used in Mamba and this will be consistent throughout the subsequent sections.

# 4 CAN MAMBA ALWAYS PERFORM COPY PERFECTLY?

In this section, we firstly interpret the reformulated SSM module introduced in Section 3 as a special linear attention. Then based on this observation, we explore the capability of Mamba to execute COPY operations during inference, which is highly related to the ability to perform in-context learning and retrieve information.

## 4.1 VIEWING MAMBA AS LINEAR ATTENTION

The attention mechanism is the key to the success of the Transformer architecture. Recent works has explored the relationship between Mamba and attention mechanisms, particularly linear attention from different perspectives Han et al. (2024); Dao & Gu (2024); Sieber et al. (2024). The linear causal attention mechanism can be formalized as:

$$\boldsymbol{y}_i = \sum_{j=1}^i \boldsymbol{v}_j \boldsymbol{k}_j^T \boldsymbol{q}_i = \sum_{j=1}^i (\boldsymbol{q}_i^T \boldsymbol{k}_j)\boldsymbol{v}_j = \sum_{j=1}^i a_{ij}\boldsymbol{v}_j, \tag{4}$$

where $\boldsymbol{q}_i, \boldsymbol{k}_i, \boldsymbol{v}_i$ are usually interpreted as query, key, value respectively and $a_{ij}$ denotes the attention scores of the $i$-th token to the $j$-th one. In attention mechanisms in Transformers, there exists $a_{ij} > 0$ for all $j \leq i$ and $\sum_{i=1}^j a_{ij} = 1$, which can be implemented by Softmax function, or approximated by kernel methods (Katharopoulos et al., 2020; Choromanski et al., 2020).

On the other hand, given the input sequence $[\boldsymbol{x}_i]_{i=1}^N$ and recalling Eq (1), that the output of the SSM module will have the following form when we set $\boldsymbol{H}_0 = \boldsymbol{O}$ and $\boldsymbol{d} = \boldsymbol{0}$:

$$\boldsymbol{y}_i = (\boldsymbol{\Delta}_i \odot \boldsymbol{x}_i)\boldsymbol{b}_i^T \boldsymbol{c}_i + \sum_{j=1}^{i-1} \left[ \boldsymbol{\Pi}_j \odot (\boldsymbol{\Delta}_j \odot \boldsymbol{x}_j)\boldsymbol{b}_j^T \right] \boldsymbol{c}_i, \tag{5}$$

where $\boldsymbol{\Pi}_j = \widetilde{\boldsymbol{A}}_i \odot \widetilde{\boldsymbol{A}}_{i-1} \odot \cdots \odot \widetilde{\boldsymbol{A}}_{j+1}$. We notice that since in practice all elements of $\boldsymbol{\Delta}$ are positive and $\boldsymbol{A}$ is set to be negative (Gu & Dao, 2023; Dao & Gu, 2024; Han et al., 2024), so that the elements of $\widetilde{\boldsymbol{A}}_i$ in Eq (1) belong to the interval $[0, 1]$. For the sake of simplicity in analysis, we replace the matrix $\widetilde{\boldsymbol{A}}_i$ with a constant $a_i$ (i.e., considering the case where all elements of $\widetilde{\boldsymbol{A}}_i$ are the same), where $a_i \in [0, 1]$ (Dao & Gu, 2024). In fact, the subsequent analysis can be easily extended to the normal case where the elements of matrix $\widetilde{\boldsymbol{A}}_i$ are different. Then, Eq (5) can be rewritten as

$$\boldsymbol{y}_i = \sum_{j=1}^i \alpha_j (\boldsymbol{\Delta}_j \odot \boldsymbol{x}_j)\boldsymbol{b}_j^T \boldsymbol{c}_i = \sum_{j=1}^i \alpha_j (\boldsymbol{c}_i^T \boldsymbol{b}_j)(\boldsymbol{\Delta}_j \odot \boldsymbol{x}_j), \tag{6}$$

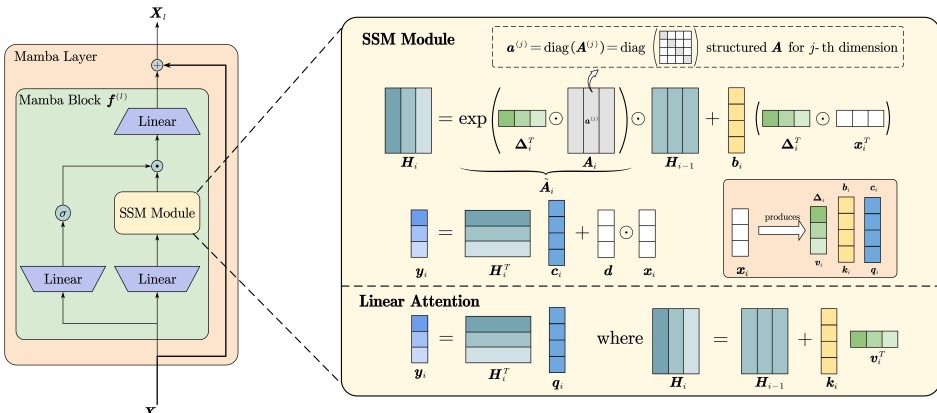

Figure 1: The illustration of the simplified Mamba layer we focus on. **Left Part**: A Mamba layer can be composed of a Mamba block with the residual connection; The Mamba block uses a gated MLP to control the output of the SSM module, where we call the branch with the SSM module as "the SSM branch" while the other as "the gated branch"; **Right Part:** The SSM module used in Mamba can be rewritten in a form similar to linear attention, where $\boldsymbol{\Delta}_i$, $\boldsymbol{b}_i$, and $\boldsymbol{c}_i$ in SSM are all derived from the current $\boldsymbol{x}_i$, similar to $\boldsymbol{v}_i$, $\boldsymbol{k}_i$, and $\boldsymbol{q}_i$ in linear attention respectively.

where $\alpha_j = \Pi_{k=j+1}^{i} a_k$ for $j \leq i-1$ and $\alpha_i = 1$. In this form, we can observe that it bears similarities to linear attention without normalization in Eq (4), where $(\boldsymbol{\Delta}_j \odot \boldsymbol{x}_j)$, $\boldsymbol{b}_j$, $\boldsymbol{c}_i$ corresponds to $\boldsymbol{v}_j$, $\boldsymbol{k}_j$ and $\boldsymbol{q}_i$ respectively and $\boldsymbol{c}_i^T \boldsymbol{b}_j$ acts like attention scores $a_{ij}$. Considering $\alpha_{j-1} \geq \alpha_j$ and $\alpha_j \in [0,1]$ for all $j \leq i$, the main difference is that each term in Eq (6) is weighted by a coefficient $\alpha_j$ to achieve the forgetting of inputs at longer distances while the attention mechanism uses the constraints for attention scores imposed by Softmax function to make sure the scaling of outputs.

## 4.2 THE TRADE-OFF OF MAMBA WHEN PERFORMING COPY

Based on the observation of the connection between the SSM module and attention mechanism, we investigate the capability of Mamba to recover historical inputs, which is foundational for the model to process information based on context. The COPY operation we focus on is defined as follows:

**Definition 1** (COPY operation). *Given the considered input sequence $\boldsymbol{X} = [\boldsymbol{x}_1]_{i=1}^{N}$, we define the $L$-local matching set $\mathcal{S}_i = \{i - L + 1 \leq j \leq i : |\boldsymbol{c}_i^T \boldsymbol{b}_j| \geq \delta\}$ and denote $\boldsymbol{v}_i = \boldsymbol{\Delta}_i \odot \boldsymbol{x}_i$ as historical records. Then the output of COPY operation is a sequence of vectors $\boldsymbol{o}_1, \boldsymbol{o}_2, ..., \boldsymbol{o}_N$ with $\boldsymbol{o}_i = \boldsymbol{v}_{pos(i)}$ where $pos(i) \in \mathcal{S}_i$ is the position we want to copy.*

The $L$-local matching set $\mathcal{S}_i$ describes the indices of historical keys $\boldsymbol{b}_j$ that is highly relevant to the current query $\boldsymbol{c}_i$ within a local window of length $L$, that is, the "attention scores $|\boldsymbol{c}_i^T \boldsymbol{b}_j|$" is lower-bounded by $\delta$. Thus, the position of the historical record we most want to replicate should be within this highly relevant set. It should be noted that here we do not specify particular conditions for the positions we want to copy for a given $i \in [N]$, which are usually determined by specific tasks. Therefore, our analysis can be applied to general scenarios. We then make the following assumption:

**Assumption 1.** *For the given sequence $\boldsymbol{X} = [\boldsymbol{x}_i]_{i=1}^{N}$, the following conditions holds:*

- *For $i \in [N]$, $\boldsymbol{c}_i^T \boldsymbol{b}_{pos(i)} \geq \rho$ where $\rho \geq \delta$ and $0 \leq |\boldsymbol{c}_i^T \boldsymbol{b}_j| < \delta$ for all $j \notin \mathcal{S}_i$.*

- *For any $i \in [N]$, there exists $\sum_{j \in \mathcal{S}_i} \alpha_j |\boldsymbol{c}_i^T \boldsymbol{b}_j| < 1$.*

The first condition of Assumption 1 places a restriction on the attention score between the historical keys and the current query: for those positions not within the $L$-local matching set $\mathcal{S}_i$, the relevance between the two will be strictly constrained within $\delta$; whereas for the position we want to replicate, its relevance is lower-bounded by $\rho$, ensuring a certain gap from other historical records. The second condition of Assumption 1 imposes a constraint on the stability of the output caused by the records in $\mathcal{S}_i$, that is, $\|\sum_{j \in \mathcal{S}_i} \alpha_j \boldsymbol{c}_i^T \boldsymbol{b}_j \boldsymbol{v}_j\| < 1$ when $\|\boldsymbol{v}_j\| \leq 1$ for $j \in \mathcal{S}_i$.

Now we present the following result:

**Theorem 1** (Perform COPY operation with constant size). *Under Assumption 1, given a input sequence $\boldsymbol{x}_1, \boldsymbol{x}_2, \ldots, \boldsymbol{x}_N$ and for any $\epsilon > 0$, there exists a Mamba block with constant size that can approximate the COPY operation, that is, for $i \in [N]$, we have $\|\boldsymbol{y}_i - \boldsymbol{o}_i\| \leq \epsilon$ if the following condition is satisfied:*

$$\rho \geq \left(1 - \frac{\epsilon}{2M\|\boldsymbol{\Delta}\|_\infty}\right)\frac{1}{\alpha_{pos(i)}} + \frac{\delta}{2}\left(\frac{a_{\max,<}}{1 - a_{\max,<}} + \frac{\left(\frac{1}{a_{\min,>}}\right)^{L-1} - 1}{1 - a_{\min,>}}\right), \qquad (7)$$

*where $a_{\max,<} = \max_{1 \leq j < pos(i)} a_j$ and $a_{\min,>} = \min_{pos(i) < j \leq i} a_j$. Moreover, in such cases, there exists $\delta \leq \frac{\epsilon}{(L-1)M\|\boldsymbol{\Delta}\|_\infty}$ when $L \geq 2$.*

The proof of Theorem 1 can be seen in Appendix A.1. Theorem 1 presents the trade-off between the historical coefficient $a_j$, the window length $L$ of $\mathcal{S}_i$, and the lower bound of the attention score $\boldsymbol{c}_i^T \boldsymbol{b}_j$ when performing the COPY operation with constant size. At first glance, the lower bound $\rho$ exhibits exponential growth with respect to $L$, that is, $\rho = \Omega\left((1/a_{\min,>})^L\right)$, which seems unacceptable.

In addition, when $a_{\min,>}$ and $L$ are smaller and $a_{\max,<}$ is larger, the aforementioned condition (7) will be satisfied more easily. More specifically, as seen from (7), when we want to copy the vector $\boldsymbol{v}_{pos(i)}$ from the hidden state, the terms before $pos(i)$ should be forgotten sufficiently ($a_{\max,<}$ is smaller) so that the record remembered by the hidden state is as pure as possible. In addition, the forgetting coefficient after $pos(i)$ should not be too small (correspondingly, $a_{\min,>}$ should be as large as possible) to make sure that $\boldsymbol{v}_{pos(i)}$ is recorded to some extent. Futhermore, the maximum distance $L$ between the copied position and the current position should not be too far. In these cases, the lower bound of the relevance score $\boldsymbol{c}_i^T \boldsymbol{b}_{pos(i)}$ will be smaller, which means that it will be easier to achieve the COPY operation. Moreover, it should be noticed that as the possible position of $pos(i)$ move further away from the current position ($L$ grows), the attention scores of irrelevant historical records will decrease, that is, $|\boldsymbol{c}_i^T \boldsymbol{b}_j|$ will be smaller for $j \notin \mathcal{S}_i$.

One cannot help but think that the sufficient condition for Mamba to achieve the COPY operation illustrated by Theorem 1 is extremely stringent as the lower bound of the attention score for the historical record we want to copy grows exponentially with $L$. This naturally leads to the question: *will Mamba perform the COPY operation more easily when we increase the model size?* We point out that when the size of Mamba increases linearly with the length of the input sequence, Mamba will be capable of accurately restoring the historical records. Below, we present our results:

**Theorem 2** (Perform COPY operation with linear-scaling size). *Given sequence $\boldsymbol{x}_1, \boldsymbol{x}_2, \ldots, \boldsymbol{x}_N \in [-M, M]^d$, there exists a Mamba block with size $O(N)$ that can perform the defined COPY operation, that is, $\boldsymbol{y}_i = \boldsymbol{o}_i$ for any $i \in [N]$. Moreover, the $l_\infty$ norm of the Mamba block parameters is upper bounded by $O(ploy(M, N))$.*

The proof of Theorem 2 can be found in Appendix A.2. Theorem 2 is based on a simple intuition: when the size grows linearly with the length of input sequence, the model will have enough space to store these historical records and therefore can retrieve them. All parameters being upper bounded by $O(ploy(M, N))$ means that the problem can also be solved by the same Mamba block with $log(N)$ precision, which has been also adopted in previous works (Merrill & Sabharwal, 2023b;a; Feng et al., 2024; Yang et al., 2024; Wen et al., 2024). Furthermore, here we only provide an existence construction, and whether the Mamba layers will actually learn these constructions is beyond the scope of our analysis. It should be noticed that during inference, such a Mamba block will have the same cost as Transformers at each step(growing linearly with length). Based on this observation, we will elaborate in Section 5 that when faced with the DP problems, Mamba will incur the same order of overhead as the Transformer.

### 4.3 MAMBA EMPIRICALLY WEAKER IN COPY TASKS THAN TRANSFORMERS

Although theoretical analysis shows that Mamba might face difficulties in performing COPY operation, in practice, Mamba may mitigate this issue through multi-layer stacking, and the actual number of parameters might be sufficient to handle tasks of a certain scale. Therefore, following previous works (Jelassi et al., 2024; Waleffe et al., 2024), we conduct experiments on both synthetic data and more realistic task to explore the practical performance of Mamba.

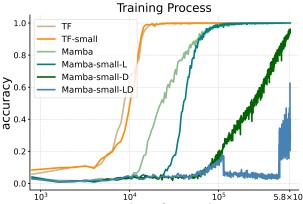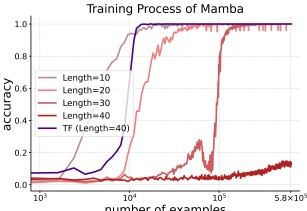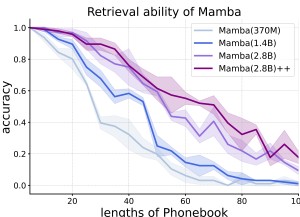

Figure 2: Experiment on the copy task and more realistic phonebook task. **Left Part:** The training process of Transformers and Mamba models of different sizes on the copy task. **Center Part:** The performance of Mamba when the length of the string to be copied is changed. **Right Part:** The performance of pre-trained Mamba as the length of the phone book increases.

### 4.3.1 EXPERIMENTS ON SYNTHETIC DATA

First, we conduct experiments on the copy task introduced by Jelassi et al. (2024). In this task, models are given a string of a certain length and then are required to copy this string exactly as output. During training, we uniformly sample a length from the interval $[N_{\min}, N_{\max}]$ and then sample characters from the alphabet at each position to obtain a string as a instance. During evaluation, we sample strings of fixed length $N_{\max}$ as inputs, and then we ask models to repeat them while counting the ratio of correctly copied characters as accuracy. More experimental details can be found in Appendix A.5.

For the left part of Figure 2, we firstly compare Transformer and Mamba of similar size (126M and 135M, labeled as TF and Mamba) while fixing the string length. The result shows that while both can perform the copy task finally, Transformer learns the task quickly whereas Mamba requires nearly seven times the number of examples. For smaller models, on the one hand, we find that the performance of Transformer is hardly affected when we halve the number of layers (63M, marked as TF-small); on the other hand, we consider two approaches for Mamba: (i) **reducing the number layers** (67M, labeled as Mamba-small-L), which causes Mamba to require more examples to achieve similar accuracy; (ii) **reducing the hidden size** (67M, labeled as Mamba-small-D), in such case Mamba learn the copy task even more slowly; (iii) **reducing the hidden size while increasing layers** (69M, labeled as Mamba-small-LD), in which case Mamba can not learn the task with $5.8 \times 10^5$ examples and the training process becomes unstable. Our results indicate that for the copy task, the hidden size of Mamba seems more important than the number of layers at the same model size, while Transformers are consistently better than Mamba.

For the center part of Figure 2, we change the maximum length $N_{\max}$ while maintaining the model size. The results showed that when $N_{max}$ are 10, 20, and 30, the number of training examples needed for Mamba to learn the task (achieving $95\%$ accuracy in 3 consecutive evaluations) is approximately $1.36 \times 10^4$, $3.52 \times 10^4$, and $1.15 \times 10^5$ respectively, indicating that the required number grows more than linearly relative to $N_{\max}$. Notably, when $N_{\max} = 40$, Mamba will be unable to learn the task within $5.8 \times 10^5$ examples. Additionally, we observe instability in Mamba's training as $N_{\max}$ increases. In contrast, Transformer can still learn quickly and maintain stability even at $N_{\max} = 40$, which again indicate that Transformer outperforms Mamba in executing copy operations.

### 4.3.2 EXPERIMENTS ON PHONEBOOK TASK

For more realistic tasks, we consider the "phonebook" task following Waleffe et al. (2024); Jelassi et al. (2024). In this task, models are given a phone book consisting of $N$ names and their corresponding phone numbers, and then models are asked in a few-shot manner to provide the phone number for some given person in the phone book, for example, "Bob: 111111; Alice: 222222; Tom: { }", which relates to the ability to copy at specified positions. We examine the pre-trained Mamba models of sizes 370M, 1.4B, and 2.8B (Gu & Dao, 2023).

It can be seen from the right part of Figure 2 that when the length $N$ of the phone book increases, the performance of models declines to a certain extent while larger models are better at retrieving the required information. Additionally, we notice that there will be some improvement in performance when we provide Mamba with task-related information in advance, such as adding the prompt "Please remember the phone number of **Tom**" at the beginning (labeled as Mamba(2.8B)++). This

suggests that informing the model about the task in advance helps it to perform better in selective memory during processing subsequent inputs, thereby enhancing the effectiveness of the copy.

# 5 THE EXPRESSIVE POWER OF MAMBA EQUIPPED WITH CoT

Although Mamba may face certain bottlenecks when handling copy tasks, another interesting question is: *when augmented with other techniques, such as Chain of Thought (CoT), will Mamba see an improvement in its capabilities?* Now, we turn our attention to the expressive ability of Mamba equipped with CoT to solve dynamic programming (DP) problems. Specifically, following the setup by Feng et al. (2024); Yang et al. (2024), a DP problem can be described by input sequences $\{s^{(1)}, s^{(2)}, \ldots, s^{(N)}\}$, state space $\mathcal{I}$, transition function $f_{\mathcal{T}}$, and aggregation function $f_{\mathcal{A}}$. Each of component can be described as follows:

• **Input sequences:** We use $\{s^{(1)}, s^{(2)}, \ldots, s^{(N)}\}$ to denote the input of the sequences and the vector $\boldsymbol{n} = \left[|s^{(1)}|, |s^{(2)}|, \ldots, |s^{(N)}|\right]^T$ to describe the scale of the problem, where $|s^{(i)}|$ denotes the length of the $i$-th sequence.

• **State Space:** For a given DP problem, the state space $\mathcal{I}_{\boldsymbol{n}}$ (and its size) will be determined based on the problem size $\boldsymbol{n}$. Each state $i \in \mathcal{I}_{\boldsymbol{n}}$ corresponds to an intermediate value $\mathrm{dp}(i)$ that needs to be computed, and $i \prec j$ means that state $i$ needs to be solved before state $j$. There exists a function $f_{\mathcal{I}} : \mathcal{I}_{\boldsymbol{n}} \to \mathcal{I}_{\boldsymbol{n}}$ to calculate the next state, that is, $j = f_{\mathcal{I}}(i)$ if $j$ is the next state to solve after $i$.

• **Transition function:** The intermediate DP value can be calculated by the transition function $f_{\mathcal{T}}$ as $\mathrm{dp}(i) = f_{\mathcal{T}}(\boldsymbol{n}, \boldsymbol{s}, \{(j, \mathrm{dp}(j)) : j \prec i\})$ where $\boldsymbol{s}$ is the concatenation of the input tokens which corresponds to all elements of all input sequences. Furthermore, this can be formulated as $\mathrm{dp}(i) = f_{\mathcal{T}}(\boldsymbol{n}, \{\boldsymbol{s}_j : j \in \mathcal{D}_i\}, \{\mathrm{dp}(k) : k \in \mathcal{V}_{\mathrm{dp}(i)}\})$ where $\mathcal{D}_i$ and $\mathcal{V}_{\mathrm{dp}(i)}$ are the sets of input tokens indices and DP values needed to solve state $i$ respectively.

• **Aggregation function:** To produce the final answer, the aggregation function needs to collect the required intermediate DP values and calculate the final result, which can be formalized as $\mathrm{anwser} = f_{\mathcal{A}}(\{\mathrm{dp}(i) : i \in \mathcal{A}_{\boldsymbol{n}}\})$ where $\mathcal{A}_{\boldsymbol{n}}$ is the set of DP values needed in the aggregation according to the problem size $\boldsymbol{n}$.

It should be noted that in the above definition, we use $s^{(i)}$ to denote the $i$-th input sequence and $\boldsymbol{s}_i$ to denote the $i$-th input token, where $\boldsymbol{s}$ is all input tokens transformed from the concatenated input sequences $(s^{(1)}, s^{(2)}, \ldots, s^{(N)})$. It can be referenced from Section 4.1 of Feng et al. (2024) for more detailed examples for DP problems. We consider the process by which the Mamba layer defined as Eq (2) gradually generates the solution to DP problems when using CoT. The format of the generated sequence can be written as:

$$s^{(1)} \mid s^{(2)} \mid \ldots \mid s^{(N)} \mid (i_1, \mathrm{dp}(i_1)) \ldots (i_{|\mathcal{I}_n|}, \mathrm{dp}(i_{|\mathcal{I}_n|})) \text{ final answer}$$

where the input sequence is separated using the symbol $\mid$ as a delimiter.

**Assumption 2.** *Given the input sequences $s^{(1)}, s^{(2)}, \ldots, s^{(N)}$, we consider the following constraints for the DP problem:*

- *For any $i \in \mathcal{I}_{\boldsymbol{n}}$, there exists $|\mathcal{D}_i| \leq N_{\boldsymbol{s}}$, $|\mathcal{V}_{\mathrm{dp}(i)}| \leq N_{\mathrm{dp}}$ and $|\mathcal{A}(\boldsymbol{n})| \leq N_{\mathcal{A}}$.*

- *The size of the state space $|\mathcal{I}_{\boldsymbol{n}}|$, all elements of input sequences (or equivalently, $\boldsymbol{s}_i$ for all $i \in [|\boldsymbol{s}|]$ ), all intermediate DP values ($\mathrm{dp}(i)$ for any $i \in \mathcal{I}_{\boldsymbol{n}}$), and the final answer can all be polynomially upper bounded by the problem size $\boldsymbol{n}$.*

- *The functions used to solve the DP problem, including the function $f_{\mathcal{I}}$ to determine the next state, the transition function $f_{\mathcal{T}}$, the aggregation function $f_{\mathcal{A}}$ and $\mathcal{A}(\boldsymbol{n})$ can all be approximated with polynomial efficiency by a constant-size MLP (with the SiLU activation function) (Feng et al., 2024).*

The first constraint of the Assumption 2 illustrates that the number of input tokens and previous DP values used in the transition function is upper bounded by $N_{\boldsymbol{s}}$ and $N_{\mathrm{dp}}$ respectively. In addition, the number of DP values used in aggregation is at most $N_{\mathcal{A}}$. The second constraint allows that all involved inputs and outputs used in functions can be represented by the log-precision model. The

third constraint allows a constant-sized degenerated Mamba (see Lemma 2) to implement functions required to solve the DP. In fact, due to the first constraint, the sizes of inputs and outputs of these functions will be a constant related to $\{N_s, N_{dp}, N_A\}$. Now we present our conclusion as follows:

**Theorem 3** (Perform DP problems with CoT). *Considering any DP problem and given input sequences that satisfies Assumption 2, for any integer $T \in \mathbb{N}$, there exists several Mamba layers with size $O(T)$, such that the answer generated by the Mamba layers will be correct when the length of the answer is no more than $T$.*

The proof of Theorem 3 can be seen in Appendix A.3. Theorem 3 states that for any DP problem, to generate correct answers with a length no greater than $T$ in CoT, the size of the Mamba layer should scale linearly with the sequence length. Additionally, it can be noted that each step

| Models with CoT | Standard TF | Sparse/Linear TF | Mamba |
|---|---|---|---|
| storage | $O(1)$ | $O(\sqrt{T})$ | $O(T)$ |
| inference each step | $O(T)$ | $O(T)$ | $O(T)$ |
| all steps | $O(T^2)$ | $O(T^2)$ | $O(T^2)$ |

Table 1: The comparison of costs for solving any DP problems between Transformers and Mamba.

of CoT for Mamba has a complexity of $O(T)$ and thus the total cost is $O(T^2)$. In comparison with standard Transformers, Feng et al. (2024) point out that a constant-sized Transformer ($O(1)$ relative to $T$) can solve any DP problem when equipped with CoT. Thus during each step of CoT inference, the inference cost for Transformer is $O(T)$, making the total cost still $O(T^2)$; While for efficient Transformers, especially linear or sparse Transformers, the conclusions obtained by Yang et al. (2024) are also similar: to solve any DP problem, they all require a hidden dimension of $O(\sqrt{T})$ (and particularly, block size $B = \Theta(\sqrt{T})$ for sparse transformers), which results in a inference cost of $O(T)$. Consequently, the total cost remains $O(T^2)$. Therefore, when equipped with CoT and solving any DP problems, Mamba does not offer additional cost savings compared to Transformers and thus all models are on equal footing in this regard. This comparison can be illustrated in Table 1.

It seems that this is a frustrating conclusion: similar to efficient Transformers, Mamba also appears not to provide savings in overhead. However, we should note that although there seems to be no shortcut to solving arbitrary DP problems, it may be possible to achieve solutions for slightly simpler problems using efficient models. We aim to show that just as efficient Transformers can offer advantages when dealing with DP problems with local properties introduced by Yang et al. (2024), Mamba can also similarly address such problems with a smaller overhead compared to standard Transformers. We assume that when solving some DP problem with CoT, the output tokens can be written as $o_1, o_2, \ldots, o_T$. If $o_i = f(\{o_j : i - m \le j < i\})$ for any $i \in [T]$, that is, the calculation for $o_i$ only depends on at most $m$ preceding intermediate DP values, then we call the DP problem is $m$-locality DP problem. With this assumption, we present the following result:

**Theorem 4** (Perform $m$-locality DP problems with CoT). *Consider any $m$-locality DP problem and given input sequences that satisfies Assumption 2, for any integer $T \in \mathbb{N}$, there exists several Mamba layers with size $O(m)$, such that the answer generated by the Mamba layers will be correct when the length of the answer is no more than $T$.*

The proof of Theorem 4 can be seen in Appendix A.4. Theorem 4 shows that when handling $m$-locality DP problem with CoT, the needed size of Mamba depends on the problem's locality. When $m$ is much smaller than the total needed length $T$, the cost for each step becomes a constant $O(m)$; therefore, the total cost becomes $O(mT)$ rather than $O(T^2)$ leading to savings in cost.

## 6 DISCUSSION

In this paper, inspired by the similarity between the SSM module in Mamba and linear attention, we explore Mamba's potential bottlenecks in the COPY operation and show that Mamba with linear size can complete it. Additionally, we present that Mamba has the same cost as standard or efficient Transformers when solving DP problems using CoT. Our findings contribute to a deeper understanding of Mamba. However, we would like to illustrate that while Mamba may slightly underperform Transformers in certain tasks, it offers advantages in others like sparse parity learning(Park et al., 2024) and can achieve comparable performance with lower costs(Gu & Dao, 2023). Therefore, as shown in Park et al. (2024); Waleffe et al. (2024); Wen et al. (2024), exploring hybrid architectures and deeper theoretical analysis for them is a promising direction for future work.

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

# A APPENDIX

## A.1 PROOF OF THEOREM 1

**Theorem 5** (Perform COPY operation). *Given a input sequence $\boldsymbol{x}_1, \boldsymbol{x}_2, \ldots, \boldsymbol{x}_N$ and for any $\epsilon > 0$, there exists an SSM module with constant size that can approximate the COPY operation defined above when the following condition is satisfied for $i \in [N]$:*

$$\rho \geq \left(1 - \frac{\epsilon}{2M\|\boldsymbol{\Delta}\|_\infty}\right)\frac{1}{\alpha_{pos(i)}} + \delta\left(\frac{a_{\max,<}}{1 - a_{\max,<}} + \frac{\left(\frac{1}{a_{\min,>}}\right)^{L-1} - 1}{1 - a_{\min,>}}\right), \tag{8}$$

*where $a_{\max,<} = \max_{1 \leq j < pos(i)} a_j$ and $a_{\min,>} = \min_{pos(i) < j \leq i} a_j$. Moreover, in such cases, there exists $\delta \leq \frac{\epsilon}{2(L-1)M\|\boldsymbol{\Delta}\|_\infty}$ when $L \geq 2$.*

*Proof.* We firstly show that given the $i$-th input $\boldsymbol{x}_i$, a SSM module can retrieve the most relevant historical record $\boldsymbol{v}_{pos(i)}$ from the hidden state and perform the defined COPY operation under the condition illustrated in our theorem. To achieve this, recalling that $\boldsymbol{v}_j = \boldsymbol{\Delta}_j \odot \boldsymbol{x}_j$ in Eq (6), we have that

$$\left\|\boldsymbol{y}_i - \boldsymbol{v}_{pos(i)}\right\|_\infty = \left\|\sum_{j=1}^i \alpha_j(\boldsymbol{\Delta}_j \odot \boldsymbol{x}_j)\boldsymbol{b}_j^T\boldsymbol{c}_i - \boldsymbol{v}_{pos(i)}\right\|_\infty = \left\|\sum_{j=1}^i \alpha_j(\boldsymbol{c}_i^T\boldsymbol{b}_j)\boldsymbol{v}_j - \boldsymbol{v}_{pos(i)}\right\|_\infty \tag{9}$$

$$= \left\|\sum_{j \neq pos(i)} \alpha_j(\boldsymbol{c}_i^T\boldsymbol{b}_j)\boldsymbol{v}_j + \alpha_{pos(i)}(\boldsymbol{c}_i^T\boldsymbol{b}_{pos(i)})\boldsymbol{v}_{pos(i)} - \boldsymbol{v}_{pos(i)}\right\|_\infty \tag{10}$$

$$= \left\|\sum_{j \neq pos(i)} \alpha_j(\boldsymbol{c}_i^T\boldsymbol{b}_j)\boldsymbol{v}_j + \left[\alpha_{pos(i)}(\boldsymbol{c}_i^T\boldsymbol{b}_{pos(i)}) - 1\right]\boldsymbol{v}_{pos(i)}\right\|_\infty \tag{11}$$

$$\leq M\|\boldsymbol{\Delta}\|_\infty\left(\sum_{j \neq pos(i)} \alpha_j|\boldsymbol{c}_i^T\boldsymbol{b}_j| + 1 - \alpha_{pos(i)}\boldsymbol{c}_i^T\boldsymbol{b}_{pos(i)}\right), \tag{12}$$

where in (12) we use the fact that $\|\boldsymbol{x}_i\|_\infty \leq M$ and $\alpha_{pos(i)}|\boldsymbol{c}_i^T\boldsymbol{b}_{pos(i)}| \leq 1$ (from the second condition in Assumption 1). Thus, to prove $\left\|\boldsymbol{y}_i - \boldsymbol{v}_{pos(i)}\right\|_\infty \leq \epsilon$, we can show that

$$\boldsymbol{c}_i^T\boldsymbol{b}_{pos(i)} \geq \left(1 - \frac{\epsilon}{M\|\boldsymbol{\Delta}\|_\infty}\right)\frac{1}{\alpha_{pos(i)}} + \sum_{j \neq pos(i)} \frac{\alpha_j}{\alpha_{pos(i)}}|\boldsymbol{c}_i^T\boldsymbol{b}_j|. \tag{13}$$

Recalling that $\alpha_j = \prod_{k=j+1}^i a_k$, we have

$$\frac{\alpha_j}{\alpha_{pos(i)}} = \begin{cases} \prod_{k=j+1}^{pos(i)} a_k = a_{pos(i)}a_{pos(i)-1}\ldots a_{j+1}, & \text{when } j < pos(i) \\ \prod_{k=pos(i)+1}^j \frac{1}{a_k} = \frac{1}{a_j a_{j-1}\ldots a_{pos(i)+1}}, & \text{when } j > pos(i). \end{cases} \tag{14}$$

Then we can consider the second term on the right side of Inequality (13) as

$$\sum_{j \neq pos(i)} \frac{\alpha_j}{\alpha_{pos(i)}}|\boldsymbol{c}_i^T\boldsymbol{b}_j| = \sum_{j \notin S_i} \frac{\alpha_j}{\alpha_{pos(i)}}|\boldsymbol{c}_i^T\boldsymbol{b}_j| + \sum_{j \in S_i, j \neq pos(i)} \frac{\alpha_j}{\alpha_{pos(i)}}|\boldsymbol{c}_i^T\boldsymbol{b}_j| \tag{15}$$

For the first term on the right side, we have

$$\sum_{j \notin S_i} \frac{\alpha_j}{\alpha_{pos(i)}} \big| \boldsymbol{c}_i^T \boldsymbol{b}_j \big| \le \delta \left( \sum_{j \notin S_i, j < pos(i)} \frac{\alpha_j}{\alpha_{pos(i)}} + \sum_{j \notin S_i, j > pos(i)} \frac{\alpha_j}{\alpha_{pos(i)}} \right) \tag{16}$$

$$\le \delta \left( \sum_{j=1}^{pos(i)-1} \frac{\alpha_j}{\alpha_{pos(i)}} + \sum_{j=pos(i)+1}^{i} \frac{\alpha_j}{\alpha_{pos(i)}} \right) \tag{17}$$

$$\le \delta \left( \sum_{k=1}^{pos(i)-1} (a_{\max,<})^k + \sum_{k=1}^{i-pos(i)} \left( \frac{1}{a_{\min,>}} \right)^k \right) \tag{18}$$

$$\le \delta \left( \frac{a_{\max,<}(1 - a_{\max,<}^{pos(i)-1})}{1 - a_{\max,<}} + \frac{\left( \frac{1}{a_{\min,>}} \right)^{i-pos(i)} - 1}{1 - a_{\min,>}} \right) \tag{19}$$

where $a_{\max,<} = \max_{1 \le j < pos(i)} a_j$ and $a_{\min,>} = \min_{pos(i) < j \le i} a_j$. In (16) we use the assumption that $\big| \boldsymbol{c}_i^T \boldsymbol{b}_j \big| le \delta$ for $j \notin S_i$; in (17) we use $\frac{\alpha_j}{\alpha_{pos(i)}} > 0$ for all $j \le i$; in (18), we use the fact that $\frac{\alpha_j}{\alpha_{pos(i)}} \le (a_{\max,<})^{pos(i)-j}$ for $j < pos(i)$ and $\frac{\alpha_j}{\alpha_{pos(i)}} \le \left( \frac{1}{a_{\min,>}} \right)^{j-pos(i)}$ for $j > pos(i)$; in (19), we use the formula for the sum of a geometric series.

Furthermore, considering that the vector $\boldsymbol{v}_{pos(i)}$ to be copied must exist in the $L$-local matching set $S_i$ so there is $i - L + 1 \le pos(i) \le i$, we have the following

$$\sum_{j \notin S_i} \frac{\alpha_j}{\alpha_{pos(i)}} \big| \boldsymbol{c}_i^T \boldsymbol{b}_j \big| \le \delta \left( \frac{a_{\max,<}(1 - a_{\max,<}^{pos(i)-1})}{1 - a_{\max,<}} + \frac{\left( \frac{1}{a_{\min,>}} \right)^{L-1} - 1}{1 - a_{\min,>}} \right) \tag{20}$$

$$\le \delta \left( \frac{a_{\max,<}}{1 - a_{\max,<}} + \frac{\left( \frac{1}{a_{\min,>}} \right)^{L-1} - 1}{1 - a_{\min,>}} \right). \tag{21}$$

In (20), we use the fact that $a_{\max,<} \in (0,1)$ while $\frac{1}{a_{\min,>}} > 1$; in (21) we just ignore the term $a_{\max,<}^{pos(i)-1}$ for simplicity (in fact, we can find that when $i$ is sufficiently large, the effect of this term can be neglected).

Meanwhile, with Assumption 1, we can consider the second term on the right side of Inequality (15) as

$$\sum_{j \in S_i, j \ne pos(i)} \frac{\alpha_j}{\alpha_{pos(i)}} \big| \boldsymbol{c}_i^T \boldsymbol{b}_j \big| \le \frac{1}{\alpha_{pos(i)}} - \boldsymbol{c}_i^T \boldsymbol{b}_{pos(i)}. \tag{22}$$

Thus, considering (13), (21) and (22), to show $\| \boldsymbol{y}_i - \boldsymbol{o}_i \|_\infty \le \epsilon$, the following condition should be satisfied

$$\boldsymbol{c}_i^T \boldsymbol{b}_{pos(i)} \ge \left( 1 - \frac{\epsilon}{M \| \boldsymbol{\Delta} \|_\infty} \right) \frac{1}{\alpha_{pos(i)}} + \delta \left( \frac{a_{\max,<}}{1 - a_{\max,<}} + \frac{\left( \frac{1}{a_{\min,>}} \right)^{L-1} - 1}{1 - a_{\min,>}} \right) \tag{23}$$

$$+ \frac{1}{\alpha_{pos(i)}} - \boldsymbol{c}_i^T \boldsymbol{b}_{pos(i)}. \tag{24}$$

After reformulating, there exists

$$\boldsymbol{c}_i^T \boldsymbol{b}_{pos(i)} \ge \left( 1 - \frac{\epsilon}{2M \| \boldsymbol{\Delta} \|_\infty} \right) \frac{1}{\alpha_{pos(i)}} + \frac{\delta}{2} \left( \frac{a_{\max,<}}{1 - a_{\max,<}} + \frac{\left( \frac{1}{a_{\min,>}} \right)^{L-1} - 1}{1 - a_{\min,>}} \right). \tag{25}$$

Recalling that $c_i b_{pos(i)}^T \geq \rho$, we can find that if we set the lower bound of $\rho$ as the right side of Inequality (25), we will have $\|y_i - o_i\| \leq \epsilon$.

Moreover, in such case, with the second condition of Assumption 1, we have $c_i^T b_{pos(i)} \leq \frac{1}{\alpha_{pos(i)}}$ thus there is

$$\frac{1}{\alpha_{pos(i)}} \geq \left(1 - \frac{\epsilon}{2M\|\Delta\|_\infty}\right) \frac{1}{\alpha_{pos(i)}} + \frac{\delta}{2} \left(\frac{a_{\max,<}}{1 - a_{\max,<}} + \frac{\left(\frac{1}{a_{\min,>}}\right)^{L-1} - 1}{1 - a_{\min,>}}\right). \quad (26)$$

By reformulating, when $L \geq 2$, we can get that

$$\delta \leq \frac{\epsilon}{M\|\Delta\|_\infty} \left(\frac{a_{\max,<}}{1 - a_{\max,<}} + \frac{\left(\frac{1}{a_{\min,>}}\right)^{L-1} - 1}{1 - a_{\min,>}}\right)^{-1} \quad (27)$$

$$\leq \frac{\epsilon}{M\|\Delta\|_\infty} \left(\frac{\left(\frac{1}{a_{\min,>}}\right)^{L-1} - 1}{1 - a_{\min,>}}\right)^{-1} \quad (28)$$

$$\leq \frac{\epsilon}{M\|\Delta\|_\infty} \cdot \frac{1}{L-1}, \quad (29)$$

where in (29) we use the inequality $\frac{1-a}{(\frac{1}{a})^x - 1} \leq \frac{1}{x}$ for $0 < a < 1$ and $x \geq 1$ where $a$ and $x$ are replaced by $a_{min,>}$ and $L-1$ in (28) respectively when $L \geq 2$.

We have shown above that an SSM module can perform the defined COPY operation under the conditions described in Theorem 1. To prove that a Mamba block can do the same, we only need to demonstrate that a Mamba block defined by Eq (3) degenerates into an SSM module. In fact, we only need to deactivate the gating branch to achieve this. For example, we can set $W_1 = W_3 = I, b_1 = 0, W_2 = O$ and $b_2 = k\mathbf{1}$ where the constant $k$ satisfies $\sigma(k) = 1$. Thus, we complete our proof. $\square$

## A.2 PROOF OF THEOREM 2

**Theorem 6** (Perform COPY operation with linear-scaling size). *Given sequence $x_1, x_2, \ldots, x_N \in [-M, M]^d$, there exists a Mamba block with size $O(N)$ that can perform the defined COPY operation, that is, $y_i = o_i$ for any $i \in [N]$. Moreover, the $l_\infty$ norm of the Mamba block parameters is upper bounded by $O(ploy(M, N))$.*

*Proof.* Recalling that the output of SSM module in Mamba can be rewritten in the form of Eq (6), where $(\Delta_j \odot x_j)$, $b_j$, $c_i$ corresponds to $v_j$, $k_j$ and $q_i$ respectively. Our intuition is to store all the information of $v_i$ from our history in the hidden state space of size $O(N)$ (similar to the KV cache in attention format), and then use the appropriate $c_i$ as the query for retrieval. We can set $A = O$ so that Eq (6) further transforms in a way that does not forget historical information, that is, $y_i = \sum_{j=1}^i (\Delta_j \odot x_j) b_j^T c_i = \sum_{j=1}^i v_j b_j^T c_i$.

Let $\tilde{x}_i = [x_i, e_i, e_{pos(i)}] \in \mathbb{R}^{d+2N}$ where $e_i \in \mathbb{R}^N$ denote the one-hot vector where only the $i$-th value is 1. We use $e_i$, $e_{pos(i)}$ to denote the current position and the position of historical token we want to copy respectively. Then, we construct $W_b = [O_{N \times d}, I_N, O_N] \in \mathbb{R}^{N \times (d+2N)}$ so that $\tilde{b}_i = W_b \tilde{x}_i = e_i$. Then, at the $i$-th step, the information newly recorded in the state space will be $v_i \tilde{b}_i^T = v_i e_i^T \in \mathbb{R}^{d \times N}$ and the updated state space will be $H_i = H_{i-1} + \sum_{j=1}^{i-1} v_j \tilde{b}_j^T + v_i \tilde{b}_i^T = [v_1, v_2, \ldots, v_i, O_{d \times (N-i)}]$ thus at the last step, we can record all historical information in the state space by $H_T = \sum_{j=1}^N v_j \tilde{b}_j^T = [v_1, v_2, \ldots, v_N]$. Then, at the output process, we can construct $W_c = [O_{N \times d}, O_N, I_N] \in \mathbb{R}^{N \times (d+2N)}$ so that $\tilde{c}_i = W_c \tilde{x}_i = e_{pos(i)}$. Thus, the output will be $y_i = H_i \tilde{c}_i = \sum_{j=1}^i v_j \tilde{b}_j^T e_{pos(i)} = v_{pos(i)}$.

At the same time, we note that in the above process, the vectors $\boldsymbol{e}_i, \boldsymbol{e}_{pos(i)}$ to denote position in $\tilde{\boldsymbol{x}}_i$ are sparse. In fact, we only need to use two indices $p_i = i$ and $p_{pos(i)} = pos(i)$ to store them thus the total size to store all indices is $O(N)$. Additionally, $\boldsymbol{W_b}, \boldsymbol{W_c}$ are also sparse so we require at most $O(N)$ space to store these two matrices. Therefore, the model size we need is $O(Nd)$ that scales linearly with the length $N$. Similar to the proof of Theorem 1, we can degenerate the Mamba block into the aforementioned SSM module by deactivating the gating branch. In addition, we note that $\boldsymbol{x}_i \in [-M, M]^d$ and $p_i, p_{pos(i)} \in [1, N]$ thus the largest value involved in the aforementioned process will not exceed $NM^2$ (the largest value in hidden states), which is upper bounded by $O(ploy(M, N))$. Thus, we complete our proof. $\qquad\square$

## A.3 PROOF OF THEOREM 3

In this part, we first present the necessary lemmas before completing the proof of Theorem 3. In fact, these lemmas are very similar to those presented by Feng et al. (2024) in Appendix C.1 regarding MLP. The main difference is that we need to degenerate Mamba blocks to MLP and consider different activation functions (SiLU for Mamba instead of GELU). Thus, we only provide detailed proofs of these relevant lemmas when necessary.

**Lemma 1** (Perform multiplication). *For any $\epsilon > 0$ and $M > 0$, there exists Mamba block parameters with $l_\infty$ norm upper bounded by $O(poly(M, 1/\epsilon))$ such that $|f(a, b) - ab| \le \epsilon$ holds for all $a, b \in [-M, M]$.*

*Proof.* We first show that a two-layer MLP using the SiLU activation function can achieve the above operation. We use the same construction as in Lemma C.1. in Feng et al. (2024), except that we use the SiLU activation function instead of GELU. Specifically, let $g : \mathbb{R}^2 \to \mathbb{R}$ be a two-layer MLP with SiLU activation, and the hidden dimension is 4, then we can construct $f$ as

$$g(a, b) = \frac{\lambda^2}{2} \left( \sigma\left(\frac{a+b}{\lambda}\right) + \sigma\left(\frac{-a-b}{\lambda}\right) - \sigma\left(\frac{a-b}{\lambda}\right) - \sigma\left(\frac{-a+b}{\lambda}\right) \right), \qquad (30)$$

where $\lambda$ is a scaling factor. In addition, considering $\sigma(x) = \frac{x}{1+e^{-x}}$, $\sigma'(x) = \frac{1+(x+1)e^{-x}}{(1+e^{-x})^2}$, $\sigma''(x) = \frac{e^{-x}(2+2e^{-x}+xe^{-x}-x)}{(1+e^{-x})^3}$, we have $\sigma(0) = 0, \sigma'(0) = \frac{1}{2}, \sigma''(0) = \frac{1}{2}$. Then, using the Taylor expansion with the Lagrange remainder, we can obtain that

$$\sigma\left(\frac{a+b}{\lambda}\right) + \sigma\left(\frac{-a-b}{\lambda}\right) - \sigma\left(\frac{a-b}{\lambda}\right) - \sigma\left(\frac{-a+b}{\lambda}\right)$$

$$= \frac{1}{2!}\frac{1}{2}\left( \left(\frac{a+b}{\lambda}\right)^2 + \left(\frac{-a-b}{\lambda}\right)^2 - \left(\frac{a-b}{\lambda}\right)^2 - \left(\frac{-a+b}{\lambda}\right)^2 \right) + R_2 = \frac{2ab}{\lambda^2} + R_2,$$

where $R_2$ is the second-order remainder term. Assuming that $\lambda > 2M$, we have $|\frac{\pm a \pm b}{\lambda}| < \frac{2M}{\lambda} < 1$ and then

$$|R_2| \le \frac{4}{3!}\left(\frac{2M}{\lambda}\right)^2 \max_{x \in [-1,1]} |\sigma'''(x)|$$

$$= \frac{4}{3!}\left(\frac{2M}{\lambda}\right)^2 \max_{x \in [-1,1]} \left| \frac{(x-3)e^{-x} - 4xe^{-2} + (x+3)e^{-3x}}{(1+e^{-x})^4} \right|$$

$$\le \frac{4}{3!}\left(\frac{2M}{\lambda}\right)^2 \frac{4e + 4e^2 + 4e^3}{(1+e^{-1})^4}$$

$$\le \frac{4}{3!}\frac{8M^3}{\lambda^3}\frac{81}{2}$$

$$= \frac{216M^3}{\lambda^3}.$$

Thus if we set $\lambda \ge \frac{216M^3}{2\epsilon}$ we will have $|g(a, b) - ab| \le \frac{\lambda^2}{2}|R_2| \le \epsilon$.

Then, we note that a Mamba block $\boldsymbol{f}$ defined as Eq (3) can degenerate into the above MLP $g$ by deactivating its SSM branch. Specifically, we only need to set $\boldsymbol{W}_1$ to be zeros and $\boldsymbol{b} = \boldsymbol{1}$ so that the

input of the SSM branch is a constant 1, that is,

$$\boldsymbol{f}(\boldsymbol{x}) = \boldsymbol{W}_3 \cdot \text{SSM}(\mathbf{1}) \odot \sigma(\boldsymbol{W}_2\boldsymbol{x} + \boldsymbol{b}_2).$$

In the SSM module, we can set $\boldsymbol{W}_b$ to be zeros, that is, no new information will be retained in the hidden state. Following this, we set $\boldsymbol{d} = \mathbf{1}$ resulting that given $\boldsymbol{x} = \mathbf{1}$, we have $\text{SSM}(\mathbf{1}) = \boldsymbol{y} = \boldsymbol{c}^T\boldsymbol{H} + \boldsymbol{d} \odot \boldsymbol{x} = \boldsymbol{c}^T\mathbf{1} + \mathbf{1} \odot \mathbf{1} = \mathbf{1}$. Thus the SSM branch can be deactivated and the Mamba block will degenerate into a two layer MLPs, that is,

$$\boldsymbol{f}(\boldsymbol{x}) = \boldsymbol{W}_3\sigma(\boldsymbol{W}_2\boldsymbol{x} + \boldsymbol{b}_2). \tag{31}$$

Furthermore, given $\boldsymbol{x} = [a, b]$, we can set $\boldsymbol{W}_2 \in \mathbb{R}^{4 \times 2}$ and $\boldsymbol{W}_3 \in \mathbb{R}^{4 \times 1}$ to meet the two-layer MLP $g$ as Eq (30). Additionally, we note that all parameters of this Mamba block can be upper bounded by $O(poly(M, 1/\epsilon))$ under the $l_\infty$ norm. Thus, we complete our proof. $\qquad\square$

**Remark 1.** *It should be noted that here we have only provided one possible construction and this is not unique. For example, in the process of deactivating the SSM branch, we could also choose to make $\tilde{\boldsymbol{\Delta}}$ sufficiently large and correspondingly $\tilde{\boldsymbol{A}}$ sufficiently small with $\boldsymbol{A} \leq \mathbf{0}$ so that the hidden states approximates zeros. In fact, the expressive power of an Mamba block with two branches should be stronger than that of a two-layer MLP since it already encompasses the latter. Nevertheless, we still provide one possible construction here.*

**Lemma 2** (Approximate two-layer MLPs with ReLU). *Let $\boldsymbol{g} : \mathbb{R}^{d_1} \to \mathbb{R}^{d_2}$ be a two-layer MLP with ReLU activation, and all parameters are upper bounded by $M$. Then, for any $\epsilon > 0$, there exists a Mamba block $\boldsymbol{f}$ and parameters upper bounded by $O(poly(M, 1/\epsilon))$ in the $l_\infty$ norm, such that for all $\boldsymbol{x} \in \mathbb{R}^{d_1}$, we have $\|\boldsymbol{f}(\boldsymbol{x}) - \boldsymbol{g}(\boldsymbol{x})\|_\infty \leq \epsilon$.*

*Proof.* Similar to Lemma 1, once again, we deactivate the SSM branch, causing a Mamba block to degenerate into the form of Eq 31. Considering a two-layer MLP with a ReLU activation function denoted as $g(\boldsymbol{x}) = \overline{\boldsymbol{W}}_3\text{ReLU}(\overline{\boldsymbol{W}}_2\boldsymbol{x})$ where $\overline{\boldsymbol{W}}_2 \in \mathbb{R}^{d \times d_1}$ and $\overline{\boldsymbol{W}}_3 \in \mathbb{R}^{d_2 \times d}$, we can set similar parameters for the degenerated Mamba blcok, that is, we consider $\boldsymbol{W}_2 = \lambda\overline{\boldsymbol{W}}_2$, $\boldsymbol{W}_3 = \frac{1}{\lambda}\overline{\boldsymbol{W}}_3$ in Eq (31) where $\lambda$ is some large constant. In order to prove the lemma, we need to show that $\|\boldsymbol{f}(\boldsymbol{x}) - \boldsymbol{g}(\boldsymbol{x})\|_\infty \leq \epsilon$ with some $\lambda$ upper bounded by $O(ploy(M, 1/\epsilon))$.

Considering a scalar $z \in \mathbb{R}$, we firstly consider the upper bound of the following equation:

$$\left|\text{ReLU}(z) - \frac{1}{\lambda}\text{SiLU}(\lambda z)\right| = \left|\max(z, 0) - \frac{z}{1 + e^{-\lambda z}}\right| = \frac{|z|}{e^{\lambda|z|} + 1} \leq \frac{1}{\lambda},$$

where we use the fact that $e^x + 1 > x$ for any $x \geq 0$. Then, let $\boldsymbol{z} = \overline{\boldsymbol{W}}_2\boldsymbol{x}$, we can show that for any $\boldsymbol{z} \in \mathbb{R}^d$,

$$\left\|\overline{\boldsymbol{W}}_3\text{ReLU}(\boldsymbol{z}) - \frac{1}{\lambda}\overline{\boldsymbol{W}}_3\text{SiLU}(\lambda\boldsymbol{z})\right\|_\infty \leq \|\overline{\boldsymbol{W}}_3\|_\infty \left\|\text{ReLU}(\boldsymbol{z}) - \frac{1}{\lambda}\text{SiLU}(\lambda\boldsymbol{z})\right\|_\infty \tag{32}$$

$$\leq Md \left\|\text{ReLU}(\boldsymbol{z}) - \frac{1}{\lambda}\text{SiLU}(\lambda\boldsymbol{z})\right\|_\infty \tag{33}$$

$$\leq Md \max_{z \in \mathbb{R}}\left|\text{ReLU}(z) - \frac{1}{\lambda}\text{SiLU}(\lambda z)\right| \tag{34}$$

$$\leq \frac{Md}{\lambda}. \tag{35}$$

Then, if we set $\lambda > \frac{Md}{\epsilon}$, we will have $\|\boldsymbol{f}(\boldsymbol{x}) - \boldsymbol{g}(\boldsymbol{x})\|_\infty \leq \epsilon$ and all parameters of the Mamba block is upper bounded by $\tilde{O}(ploy(M, 1/\epsilon))$. Thus, we complete our proof. $\qquad\square$

**Remark 2.** *We have proven that a Mamba block can approximate a two-layer MLP with ReLU activation function, and since the latter can perform many basic operations, including linear transformations and selection operations as constructed in Lemma C.3 and Lemma C.5 in Feng et al. (2024), we can use Lemma 2 to adopt the same construction, enabling the Mamba block to perform these operations. We present the following colloary more specifically, and the detailed proof can be found in the above mentioned part in Feng et al. (2024).*

**Lemma 3** (Perform linear transformation, easily derived from Lemma 2 and Lemma C.3 in Feng et al. (2024)). *Let $\boldsymbol{W} \in \mathbb{R}^{d_2 \times d_1}$ be any matrix used for implementing linear transformations upper bounded by $M$ and $\boldsymbol{f} : \mathbb{R}^{d_1} \to \mathbb{R}^{d_2}$ be a Mamba block. Then, for any $\epsilon > 0$, there exist Mamba block parameters with $l_\infty$ norm bounded by $O(poly(M, 1/\epsilon))$, such that for any $\boldsymbol{x} \in \mathbb{R}^{d_1}$, we have $\|\boldsymbol{f}(\boldsymbol{x}) - \boldsymbol{W}\boldsymbol{x}\|_\infty \leq \epsilon$.*

**Lemma 4** (Perform select operation, easily derived from Lemma 2 and Lemma C.4 in Feng et al. (2024)). *Define the selection function $\boldsymbol{g} : \mathbb{R}^d \times \mathbb{R}^d \times \mathbb{R} \to \mathbb{R}^d$ as follows:*

$$g(\boldsymbol{x}, \boldsymbol{y}, t) = \begin{cases} \boldsymbol{x} & if \ t > 0 \\ \boldsymbol{y} & if \ t < 0 \end{cases} \tag{36}$$

*Let $\boldsymbol{f} : \mathbb{R}^d \times \mathbb{R}^d \times \mathbb{R} \to \mathbb{R}^d$ be a Mamba block. Then, for any $\epsilon > 0$, $\alpha > 0$, and $M > 0$, there exist Mamba parameters with $l_\infty$ norm bounded by $O(poly(M, 1/\alpha, 1/\epsilon))$, such that for all $\boldsymbol{x} \in [-M, M]^d$, $\boldsymbol{y} \in [-M, M]^d$, and $t \in [-\infty, -\alpha] \cup [\alpha, +\infty]$, we have $\|\boldsymbol{f}(\boldsymbol{x}, \boldsymbol{y}, t) - g(\boldsymbol{x}, \boldsymbol{y}, t)\|_\infty \leq \epsilon$.*

Next, we show that one Mamba layer or several Mamba layers can implement indicator functions through the select operation. We mainly focus on the usual indicator functions $\mathbb{I}[a \neq b]$, $\mathbb{I}[a > b]$ and $\mathbb{I}[a < b]$.

**Lemma 5** (Perform indicator function). *Define the indicator function $\mathbb{I}(a, b, \circ) : \mathbb{R}^2 \times \{\neq, >, <\} \to \{0, 1\}$ where $a, b \in [-M, M]$. The output of the function will be 1 if $a \circ b$ is satisfied otherwise the output will be 0. Let $f : \mathbb{R}^2 \to \mathbb{R}$ be a Mamba block. Then, for any $\epsilon > 0$, there exist Mamba parameters with $l_\infty$ norm upper bounded by $O(poly(M, 1/\epsilon))$, such that for any $a, b \in [-M, M]$ and $\circ \in \{\neq, >, <\}$, we have $\|f(a, b) - \mathbb{I}(a, b, \circ)\|_\infty \leq \epsilon$.*

*Proof.* We first show that a Mamba block can implement $\mathbb{I}[a > b]$ and $\mathbb{I}[a < b]$. For $\mathbb{I}[a > b]$, it is equivalent to consider $g(1, 0, a - b)$ where $g(\cdot)$ defined in Lemma 4. So firstly we can use a linear layer with appropriate parameters $\boldsymbol{W}_0, \boldsymbol{b}_0$ to convert the input $[a, b]$ into the vector $[1, 0, a - b]$. Then we can use Lemma 4 to implement $\mathbb{I}[a > b]$ by changing the parameters of the first linear layer from $\{\boldsymbol{W}_1, \boldsymbol{b}_1\}$ to $\{\boldsymbol{W}_1\boldsymbol{W}_0, \boldsymbol{b}_1 + \boldsymbol{W}_1\boldsymbol{b}_0\}$. The proof for $\mathbb{I}[a < b]$ is similar as well.

Noticing that $\mathbb{I}[a \neq b] = 1 - (1 - \mathbb{I}[a > b]) \cdot (1 - \mathbb{I}[a < b])$, we can implement $\mathbb{I}[a \neq b]$ through the following layers: Firstly, we can use one Mamba block to implement $1 - \mathbb{I}[a > b]$ and $1 - \mathbb{I}[a < b]$ simultaneously, where the hidden dimension will be 8 and the output is a vector $[1 - \mathbb{I}[a > b], 1 - \mathbb{I}[a < b]]$. Then, another Mamba block is constructed to implement the multiplication $(1 - \mathbb{I}[a > b]) \cdot (1 - \mathbb{I}[a < b])$ according to Lemma 1 and the appropriate outermost linear layer parameters are chosen to simultaneously achieve multiplication by a negative sign and addition of a bias of 1, where the hidden dimension will be 4 and the output will be $\mathbb{I}[a \neq b]$. Thus, we complete our proof. $\qquad \square$

Now, based on the basic operations that can be implemented by the Mamba blocks as discussed above, we present the proof of Theorem 3:

**Theorem 7** (Perform DP problems with CoT). *Considering any DP problem and given input sequences that satisfies Assumption 2, for any integer $T \in \mathbb{N}$, there exists several Mamba layers with size $O(T)$, such that the answer generated by the Mamba layers will be correct when the length of the answer is no more than $T$.*

*Proof.* Firstly, we illustrate the input format for the DP problem. We follow the embedding format in the proof of Theorem 4.7 in Feng et al. (2024), that is, assuming that the input at any step of solving the DP problem using CoT is a sequence of tokens embedded as follows:

$$\boldsymbol{x}_t^{(0)} = \left[ \boldsymbol{e}_t^{\text{input}}, \boldsymbol{e}_t^{\text{state}}, \boldsymbol{e}_t^{\text{dp}}, \boldsymbol{e}_t^{\text{answer}}, \boldsymbol{e}_t^{\text{sep}}, t, 1 \right],$$

where the specific value of each part is depend on the content represented by the current token. More specifically, each part can be described as:

- If the current position denotes a input token, then we set $\boldsymbol{e}_t^{\text{input}}$ as the embedding of the input and simultaneously set $\boldsymbol{e}_t^{\text{state}} = \boldsymbol{e}_t^{\text{dp}} = \boldsymbol{e}_t^{\text{answer}} = \boldsymbol{e}_t^{\text{sep}} = \boldsymbol{0}$.

- If the current position is the final answer, then $e_t^{\text{anwser}}$ denotes the embedding of the answer and we set $e_t^{\text{input}} = e_t^{\text{state}} = e_t^{\text{dp}} = e_t^{\text{sep}} = \mathbf{0}$.

- If the current position denotes the $j$-th separator | between input sequences , then we set $e_t^{\text{sep}} = e_j$ and $e_t^{\text{input}} = e_t^{\text{state}} = e_t^{\text{dp}} = e_t^{\text{answer}} = \mathbf{0}$.

- If the current position denotes an intermediate DP state, then we use $e_t^{\text{state}}$ to denote the embedding of the DP state and $e_t^{\text{dp}}$ denotes the corresponding value. Similarly, other part will be set to be $\mathbf{0}$.

- The scalar $t$ denotes the current position in the whole sequence, which holds the value for all above cases.

We illustrate that here we use a concatenation operation to replace the residual connection definced in Eq (2), which is a technique also used by Feng et al. (2024); Yang et al. (2024) in similar proofs. This is because, from the perspective of expressive capability, the two operations are equivalent: the output of a Mamba block $y = f(x)$ concatenated with the input (that is, $[y, x]$) can also be represented using the residual connection: $g([x, \mathbf{0}]) + [\mathbf{0}, x] = [y, \mathbf{0}] + [\mathbf{0}, x] = [y, x]$ where $g : \mathbb{R}^{2d} \to \mathbb{R}^{2d}$ is another Mamba block and part of its parameters to be same as $f$ and others are set to be $\mathbf{0}$. Conversely, the concatenation can implement residual connection by using a linear projection.

Here, we show our construction of several Mamba layers to solve the DP problem, which is composed of different blocks to perform different tasks:

**Block 1:** The first block aims to calculate the problem size $n$ and the embedding of the next state $e_t^{\text{next\_state}}$. This process can be described as follows:

- **Compute the problem size $n$:** (i) First, we can replicate the position of the token $t_{sep,1}, t_{sep,2}, \ldots, t_{sep,N}$ using the COPY operation. This can be achieved with a Mamba layer of size $O(Ntd)$ according to Theorem 2; (ii) Then, we calculate the size of the problem as $n = [t_{\text{sep},1} - 1, t_{\text{sep},2} - t_{\text{sep},1} - 1, \ldots, t_{\text{sep},N} - t_{\text{sep},N-1} - 1]$, which can be done by applying a linear transformation using one Mamba layer, as shown by Lemma 3.

- **Obtain the next state $e^{\text{next\_state}}$:** According to Assumption 2, the function $e^{\text{next\_state}} = f(n, e^{\text{state}})$ which determines the next state, can be approximated by constant-sized MLPs. Thus, this can also be implemented by having several Mamba layers degenerate into MLPs.

The output after this step can be written as:

$$x_t^{(1)} = [e_t^{\text{input}}, e_t^{\text{state}}, e^{\text{next\_state}}, e_t^{\text{dp}}, e_t^{\text{answer}}, e_t^{\text{sep}}, n, t, 1]$$

**Block 2:** The second block is mainly constructed to find the indices of input tokens and intermediate DP values that are needed to calculate the DP value corresponding to $e^{\text{next\_state}}$. Specifically, this can be described as follows:

- **Calculate the needed indices:** We calculate the positions of the input token $p_t^s = I_s(n, e^{\text{state}})$ and the positions of tokens that correspond to needed DP values $p_t^{\text{dp}} = I_{\text{dp}}(n, e^{\text{state}})$. If $I_s(n, e^{\text{state}}) = \emptyset$ or $I_{\text{dp}}(n, e^{\text{state}}) = \emptyset$, we set the positions to be a special value $\gamma$. According to Assumption 2, these two functions can be done by constant-size MLPs thus can be approximated by degenerated Mamba layers.

- **Set the flag:** (i) Set the flag $f_t^{\text{answer}}$ based on whether the DP value of the current state is needed in the final aggregation function. This can be achieved by several Mamba layers with Assumption 2 that the function $\mathcal{A} = f(n, s)$ can be approximated by MLPs and additionally using Lemma 5 to implement $\mathcal{I}[e_t^{\text{state}} \neq e_j^{\text{state}}]$ where $e_j^{\text{state}} \in \mathcal{A}$. (ii) Set the flag $f_t^{\text{state}}$ to denote whether the current state is the last state. This can be implemented by checking $\mathbb{I}[e_t^{\text{next\_state}} \neq \mathbf{0}]$ with Mamba layers using Lemma 5.

The output result after this step can be written as:

$$\boldsymbol{x}_t^{(2)} = [\boldsymbol{e}_t^{\text{input}}, \boldsymbol{e}_t^{\text{state}}, \boldsymbol{e}^{\text{next\_state}}, \boldsymbol{e}_t^{\text{dp}}, \boldsymbol{e}_t^{\text{answer}}, \boldsymbol{e}_t^{\text{sep}}, \boldsymbol{n}, \boldsymbol{p}_t^{\boldsymbol{s}}, \boldsymbol{p}_t^{\text{dp}}, f_t^{\text{answer}}, f_t^{\text{state}}, t, 1]$$

**Block 3:** This block is designed to calculate the DP value for the next state. In detail, the implementation involves the following steps:

- **Check the flag:** We check the flag $f_t^{\text{state}}$ using several Mamba layers using 5 to implement $\mathbb{I}[f_t^{\text{state}} \neq 1]$ using Lemma 5. If $f_t^{\text{state}} = 1$, the current denote is the last state and we just need to set $\boldsymbol{p}_t = \gamma \boldsymbol{1}$ where $\boldsymbol{p}_t$ denotes $\boldsymbol{p}_t^{\boldsymbol{s}}$ and $\boldsymbol{p}_t^{\text{dp}}$, which implies $I_{\boldsymbol{s}}(\boldsymbol{n}, \boldsymbol{e}^{\text{state}}) = \emptyset$ and $I_{\text{dp}}(\boldsymbol{n}, \boldsymbol{e}^{\text{state}}) = \emptyset$, that is, no input tokens or DP values are needed.

- **Obtain the needed embeddings:** If $f_t^{\text{state}} \neq 1$, then (i) We copy the input token embeddings $\boldsymbol{e}^{\text{input}}$ at positions $\boldsymbol{p}_t^{\boldsymbol{s}}$. This COPY operation can be implemented by a Mamba layer of size $O(N_s td)$ using Theorem 2; (ii) Simultaneously, we copy the embeddings of DP values at positions $\boldsymbol{p}_t^{\text{dp}}$, which can be achieved by a Mamba layer of size $O(N_{\text{dp}} td)$. If the position is empty, we just need to check $\mathbb{I}[\boldsymbol{p}_t \neq \gamma \boldsymbol{1}]$ and set the needed embeddings $\boldsymbol{e}^{\text{input}}$ or $\boldsymbol{e}^{\text{dp}}$ to be some special token. Totally, the size of Mamba layers in this step is $O((N_s + N_{\text{dp}})td)$.

- **Calculate the DP value:** We calculate the DP value $\boldsymbol{e}_t^{\text{next\_state}}$ for the next state with the Assumption 2 that the transition function can be approximated by several Mamba layers using Lemma 2.

The output result after this step can be written as:

$$\boldsymbol{x}_t^{(2)} = [\boldsymbol{e}_t^{\text{input}}, \boldsymbol{e}^{\text{next\_state}}, \boldsymbol{e}_t^{\text{next\_dp}}, \boldsymbol{e}_t^{\text{answer}}, \boldsymbol{e}_t^{\text{sep}}, \boldsymbol{n}, f_t^{\text{answer}}, f_t^{\text{state}}, t, 1]$$

**Block 4:** The last block is constructed to implement the final aggregation function and output the final answer. Specifically, the steps are as follows:

- **Check the flag:** We identify whether the current state is the last state by checking $\mathbb{I}[f_t^{\text{state}} \neq 1]$ by using Lemma 5. If $f_t^{\text{state}} = 1$, then all intermediate DP values have been solved and we need to compute the final answer.

- **Obtain the needed embeddings:** We collect the DP value embeddings $\boldsymbol{e}^{\text{dp}}$ of these tokens whose $f^{\text{answer}} = 1$, which can be achieved by COPY operation according to Theorem 2 with one Mamba layer of size $O(N_{\mathcal{A}} td)$.

- **Generate the final answer:** Finally, we compute the answer by implementing the aggregation function, which can be achieved by constant-size MLPs according to Assumption 2, thus can also be achieved by several degenerated Mamba layers.

In summary, given a sequence length $t$ and equipped with CoT, the parameter size required by the Mamba layers to generate the correct answer at each step is $O(\tilde{N}td)$, where $\tilde{N} = \max\{N, N_{\boldsymbol{s}} + N_{\text{dp}}, N_{\mathcal{A}}\}$ is a constant independent of $t$, that is, the size of the Mamba layer scales linearly with $t$. Thus, we complete our proof. $\square$

A.4 PROOF OF THEOREM 4

**Theorem 8** (Perform $m$-locality DP problems with CoT). *Consider any $m$-locality DP problem and given input sequences that satisfies Assumption 2, for any integer $T \in \mathbb{N}$, there exists several Mamba layers with size $O(m)$, such that the answer generated by the Mamba layers will be correct when the length of the answer is no more than $T$.*

*Proof.* The overall proof construction approach is similar to that of Theorem 3, with the only difference being that under the assumption of $m$-locality, when performing the COPY operation, the constructed Mamba only needs to focus on at most $m$ tokens preceding the current position. This results in the size of the Mamba layers only needing to be $O(\tilde{N}md)$. $\square$

## A.5  MORE DETAILS OF EXPERIMENTS

For the copy task experiments, we mainly refer to the setup by Jelassi et al. (2024). For Transformers, we select the GPT-NeoX architecture (Andonian et al., 2023) while for Mamba we use the Mamba GitHub repository(Gu & Dao, 2023) and all experiments are conducted on A800 GPUs.

More specifically, for the left part of Figure 2, we configure 10 layers for the Transformer (TF) and 5 layers for TF-small, with both having a hidden size of 1024 and RoPE (Su et al., 2024) as the positional encoding. For Mamba models, we configured 20 layers and a hidden size of 1024 for Mamba, 20 layers and a hidden size of 720 for Mamba-small-D, 10 layers and a hidden size of 1024 for Mamba-small-L, and 40 layers and a hidden size of 512 for Mamba-small-LD. We use an online sampling batch size of 8 and set the maximum context length to 220, meaning each example often contains multiple instances. AdamW(Loshchilov, 2017) is chosen as the optimizer with a learning rate of 1e-5 and weight decay of 0.1. We set $N_{min} = 10$ and $N_max = 30$ for all models.

For the middle part of Figure 2, the Transformer and Mamba setups match the aforementioned configurations for TF and Mamba. Moreover, we set $[N_{min}, N_{max}]$ to $[5, 10]$, $[10, 20]$, $[20, 30]$ and $[30, 40]$ for sequence lengths of 10, 20, 30 and 40 respectively.

For the right part of Figure 2, we use pretrained Mamba models of size 370M, 1.4B and 2.8B(Gu & Dao, 2023), which are pretrained on the Pile(Gao et al., 2020). For Mamba(2.8B)++, the prompt at the begining is just like:

"The following is a phonebook with the form: Gary Battle: 8444797678 Gary Gallegos: 9960330831.

Remeber the phone number of Joseph Perry. Here is the phonebook:..."

