# OpenReview forum: "Can Mamba Always Enjoy the "Free Lunch"?"
_ICLR.cc/2025/Conference — Submitted to ICLR 2025_

### Official Review · Reviewer_vUfm · 2024-11-03

**Soundness:** 2
**Presentation:** 2
**Contribution:** 2
**Rating:** 5
**Confidence:** 4

**Summary:**

This paper provides theoretical analysis to the question of whether Mamba can always enjoy the “free lunch”, that is whether it is able to achieve similar performance as Transformer while remaining significantly lower cost. The paper provides analysis for two tasks: COPY and dynamic programming  (DP), and demonstrates that for the COPY task, fixed-size Mamba may encounter bottlenecks, unless the size of the model grows linearly with sequence length, and for the DP task there is no particular advantage of Mamba unless the problem has a local structure.

**Strengths:**

This paper provides some in-depth theoretical study which elucidates the limits of the Mamba model architecture. These contributions help researchers better understand the trade-off between performance and computational cost for the two popular model architectures.

**Weaknesses:**

Existing works already demonstrated similar limitations of Mamba (and SSM in general) in handling tasks like COPY (especially [1,2], which the paper also referred to). The conclusions of this paper did not appear to have provided much additional insight, and the contributions are incremental. What is more, the theoretical derivation and results of the paper (Theorem 1-4, and the assumptions they build upon) seem to be pure complex mathematical constructions, which do not appear to be concise and interpretable enough to be fully convincing or empirically verifiable.

[1] Repeat After Me: Transformers are Better than State Space Models at Copying
[2] The Illusion of State in State-Space Models

**Questions:**

- The theoretical result in Theorem 1 is honestly a bit difficult to understand intuitively: Is this a loose or tight bound? It appears to be a characterization of the difficulty of a COPY problem, but it is difficult to get a sense of what it indicates in practice as there are many variables introduced to control this bound. It would be great to justify the interpretability of this theoretical result and how this bound is an accurate characterization of the model behavior (e.g. plot curves comparing theoretical bounds with empirical results).

- Similarly, the analysis of the paper depends on critical Assumptions 1 and 2. While I understand these are mathematical constructions needed to make the derivations go through, these assumptions appear to be a bit too convoluted and difficult be measured or justified empirically. In general, there are many ways to derive theoretical upper and lower bounds, depending on the choice of assumptions, however an “optimal” theory should be based on minimal assumptions while producing interpretable outcomes.

- It would be great to also provide some experimental examples to support the claim of Sec. 5 (Mamba does not offer additional cost savings compared to Transformers for DP problems), similar to the COPY problem.

- In Fig. 2, what is the Transformer baseline in the rightmost chart?

---

> ### Author Response · Authors · 2024-11-20
>
> We greatly appreciate the reviewer's careful review and constructive suggestions. Below are our responses.
>
> > **W1:** Existing works already demonstrated similar limitations of Mamba (and SSM in general) in handling tasks like COPY (especially [1,2], which the paper also referred to). The conclusions of this paper did not appear to have provided much additional insight, and the contributions are incremental.
>
> The tasks we focus on are fundamentally different: Merrill et al. (2024) [1] emphasize that Mamba is also unable to solve **state tracking**; Jalessi et al. (2024) [2] concentrate on scaling in terms of **sequence length** and the number of states. In contrast, our work examines the impact of the distance to the target token on output error from the perspective of **numerical approximation**. Additionally, our **underlying intuition** is distinct: our analysis is inspired by the relationship between linear attention and Mamba, which is an aspect not addressed in the two aforementioned works.
>
> [1] The Illusion of State in State-Space Models
>
> [2] Repeat After Me: Transformers are Better than State Space Models at Copying
>
> > **W2:** What is more, the theoretical derivation and results of the paper (Theorem 1-4, and the assumptions they build upon) seem to be pure complex mathematical constructions, which do not appear to be concise and interpretable enough to be fully convincing or empirically verifiable.
>
> Intuitively, in Theorem 1, we provided a sufficient condition for Mamba to perform the COPY operation, while Theorems 2 to 4 aim to establish an upper bound on the model size required by Mamba blocks to solve the target tasks. That is, they show the maximum model size needed to accomplish COPY/DP tasks.
>
> In addition, as highlighted by reviewer `k7wk`, we will include additional examples in future versions to make the implications of these conclusions more intuitive (you can also see our response to **Q2**). Furthermore, similar to the theoretical analysis in prior works, our theoretical section focuses on the expressive power of the Mamba block, rather than whether this ability can be learned. However, in Section 4.3, we provide some experiments to demonstrate phenomena related to these theoretical results.
>
> > **Q1:** The theoretical result in Theorem 1 is honestly a bit difficult to understand intuitively: Is this a loose or tight bound? It appears to be a characterization of the difficulty of a COPY problem, but it is difficult to get a sense of what it indicates in practice as there are many variables introduced to control this bound. It would be great to justify the interpretability of this theoretical result and how this bound is an accurate characterization of the model behavior (e.g. plot curves comparing theoretical bounds with empirical results).
>
> This explanation of Theorem 1 might make it easier to understand: for constant-size Mamba blocks, we identified the **sufficient conditions** under which Mamba can perform the COPY operation. Rather than emphasizing whether these conditions form a tight or loose bound, we prefer to describe them as stringent. This is stringent because we are considering a fairly general scenario involving a given sequence $X = [x_i]_{i=1}^{N}$ and positions $pos(i)$.  Of course, Mamba blocks may still be able to perform the COPY operation in less restrictive cases where the sufficient conditions are not satisfied. That is, given any sequence and position, it remains unclear whether constant-size Mamba blocks can always perform the COPY operation. However, we want to point out that Mamba blocks of size $O(N)$ **always can.**

---

> ### Author Response · Authors · 2024-11-20
>
> > **Q2:** Similarly, the analysis of the paper depends on critical Assumptions 1 and 2. While I understand these are mathematical constructions needed to make the derivations go through, these assumptions appear to be a bit too convoluted and difficult be measured or justified empirically. In general, there are many ways to derive theoretical upper and lower bounds, depending on the choice of assumptions, however an “optimal” theory should be based on minimal assumptions while producing interpretable outcomes.
>
> First, regarding Assumption 1: for the first condition, the appearance of $\rho$ and $\delta$ serves primarily as a convenient definition for expressing the theorem. We do not impose strict restrictions on the values of $\rho$ and $\delta$. Their sizes are flexible, but they generally form a trade-off with the conditions in Theorem 1; the second condition is simply to ensure the stability of the output scale. This assumption is necessary; otherwise, during theoretical analysis, the output scale could increase significantly as $|S_i|$ grows. In practice, this issue can be mitigated by using modules like layer normalization. However, incorporating such mechanisms into theoretical analysis would introduce additional complications, which is why we opted for the current assumption.
>
> We would like to provide an example:  Given a text snippet, "XXXXXXX(Some unrelated content). Bob has a dog while John has a cat. After school, Bob often plays with his ____." We assume the expected output from the model here is "dog". In this example, assuming its position of the answer is $i$, the Matching Set $S_i$ for the current token to be predicted and $L \approx 18$ represents the distance within which the answer might exist relative to position $i$. In other words, starting from the phrase "Bob has a ...". In addition, $pos(i)$ refers to the position index of the word "dog".  The Matching Set $S_i$ might include position indices for words like "Bob," "dog," "John," and "cat" if we choose some $\delta$. Of course, if we choose a larger $\delta$, the Matching Set $S_i$ might only include the position indices corresponding to "Bob" and "dog".
>
> For Assumption 2:  A significant number of DP problems conform to such conditions. For example, consider the problem of finding the minimum edit distance, where we aim to transform string $s_1$ into string $s_2$ with lengths $n_1 = |s_1|$ and $n_2 = |s_2|$, respectively. The costs for insertion, deletion, and substitution are $a$, $b$, and $c$, respectively. Let $\rm dp(j,k)$ represent the cost of transforming the first $j$ characters of $s_1$ into the first $k$ characters of $s_2$. The transition function can then be expressed as:
>
> $$
> \text{dp}(j, k) =\begin{cases}    ak & \text{if } j = 0 \\\\    bj & \text{if } k = 0 \\\\    \min \big(         \text{dp}(j, k-1) + a, \text{dp}(j-1, k) + b,\\\\        \qquad \text{dp}(j-1, k-1) + c \mathbb{I}[s_j^{(1)} \neq s_k^{(2)}]    \big) & \text{otherwise}\end{cases}
> $$
> Finally, the aggregation function selects $\rm dp(n_1, n_2)$ as the final answer.  In the example above, the size of the state space, all intermediate values, and the lengths of the input strings will all be upper-bounded by $\rm poly(n_1, n_2)$. Moreover, the operations required by the involved functions can be approximated with polynomial efficiency by a constant-size MLP, as shown in Lemmas 1-5. Therefore, such DP problems satisfy Assumption 2.
>
> > **Q3:** It would be great to also provide some experimental examples to support the claim of Sec. 5 (Mamba does not offer additional cost savings compared to Transformers for DP problems), similar to the COPY problem.
>
> Thank you for your suggestion. We will design CoT experiments in the future revisions to support our claim.
>
> > **Q4:** In Fig. 2, what is the Transformer baseline in the rightmost chart?
>
> In Fig. 2, as mentioned in Section 4.3.2, we did not compare against Transformer; instead, we only compared different sizes of pre-trained Mamba blocks.

---

> ### Comment · Reviewer_vUfm · 2024-12-03
>
> I'd like to thank authors for your responses. I would like to keep my original rating.

---

### Official Review · Reviewer_k7wk · 2024-11-04

**Soundness:** 3
**Presentation:** 2
**Contribution:** 4
**Rating:** 5
**Confidence:** 3

**Summary:**

The paper provides a theoretical overview of Mamba by following a reformulation to make a direct comparison to attention in Transformers. Then, the authors present theorems in the context of a “COPY” task (akin to correctly outputting a token from an earlier position). First, they prove that Mamba can indeed perform the COPY task; next, they prove that performing COPY for any token from a length-N sequence would require O(N)-sized Mamba blocks. Empirical results on a small synthetic dataset also show that Mamba still performs worse than Transformers on training dynamics, but otherwise correspond with the proven results. Finally, the authors explore chain-of-thought solutions for dynamics programming (DP) problems which leverage past results, proving that a sufficiently-sized Mamba can generate the correct answer and that actually depending on the type of DP problem, a tighter bound on Mamba size can be found.

**Strengths:**

1. The main results (Theorems 1 and 2) on the relation between token/recall distance and size of the Mamba block, while overlapping with Jelassi et al., 2024, are still different and appear new.

2. There is insightful empirical analysis on learning dynamics for Mamba (for COPY synthetic task) that complement the theoretical results.

3. Additional theoretical results (Theorems 3 and 4) relate the more general conclusion from Theorems 1 and 2 to an applied problem setting (DP problems), including a comparison against standard and sparse transformers. The proof/extension to DP/CoT-type problems in particular is nontrivial, although is in the appendix presumably due to space constraints.

**Weaknesses:**

1. While the inclusion of DP problems feels necessary to relate some of the abstract findings to a real problem (CoT with DP), the findings feel inconclusive because it teaches us about representational power and is an existence proof. However, as we observe in the Phonebook experiments, that doesn’t translate in practice to viability; in particular the bounds are still quite loose. When reading the paper, it felt like Section 5 was a separate paper as it introduces a new set of notation despite the result being ultimately a corollary of the original theorem.

2. It isn’t stated clearly why the assumptions (both Assumption 1 and Assumption 2) are safe to assume. When are these assumptions satisfied?

Assumption 1, first condition – I don’t understand why it is reasonable to assume this. Assume $L=1$, and then we would need $|c_i^\top b_{pos(i)}|$ to be maximal for all $i$. This doesn't seem to be necessarily true for all tokens, like stop words. While this feels more likely true as we increase $L$, it still doesn't feel like a safe assumption. Even the assumption that for large $L$, all $|c_i^\top b_j| > \delta$ for all $j \in S_i$ seems unlikely as there could be one token that ends up with a low score. I guess I don’t understand why we think $|c_i^\top b_j|>\rho$ just because $j \in S_i$. Separately, if we want to ensure a “gap”, shouldn’t there be an $\epsilon > 0 $ so that  $\rho \ge \delta + \epsilon$?

Assumption 1, second condition - this is missing a something. “There exists $S_i$ such that…” or “There exists $j$ such that?

For all the conditions of Assumption 2, it would be helpful to give a title to each condition, like “Existence of expressible solution”, “DP is poly(n)” and “DP steps can be approximated by Mamba”.

3. The presentation of the core theorems (Section 4.2) is very difficult to understand. It might be easier to present Theorem 1 in terms of $\delta$ (and use $\rho$ only in the proof - the first condition doesn't actually need $\rho$ either). For Theorem 2, the “simple intuition” could be better expressed in relation to the terms of Eq. 7, i.e. as a direct result of the first term increasing.

Some other examples:
$M$ is not defined in equation 7, but it should be mentioned as $M = \max_i||x_i||_\infty$; as mentioned above, the 2nd condition of Assumption 1 is unclear’ “ploy” -> “poly”; we should be targeting output tokens, but it seems like the problem statement has changed to target vectors.


**In summary** of 1, 2, 3, combining the fact that the assumptions are not obvious or intuitive, the conditions of Eq 7 seems possibly difficult to satisfy, and the middling empirical results, it is hard to know whether the existence proof is a rare special case or something more in general. It is still interesting even if it is a rare special case as it is indicative of representation/expressivity of Mamba, but the lack of discussion around it makes its impact hard to understand.


4. Finally, the empirical results would be significantly more interpretable if instead of the total number of parameters, the actual values of the Mamba block size were reported, since that's what the theoretical results are in reference to.

**Questions:**

1. “free lunch” is not well-defined, and using it in the title is a bit misleading, since it is specifically referring to the sequence length/COPY task. My one-sentence understanding of all the findings is:

> If the distance of the token or information being recalled is N, then Mamba needs block size O(N) to recall that information.

Do you agree with this generalized statement? It would be an intuitive corollary (which you also have the proof for) then that for an m-local DP problem with T steps, it would take O(m) size and O(mT) cost. I want to make sure I am not misunderstanding the paper.

2. What’s the significance of the $l_\infty$ norm in Theorem 2? Is it a corollary, assumption, or not actually relevant to the results?

3. Is it possible to express the DP conditions of Assumption 2 in terms of a formal problem complexity class? It reminds me of DSPACE(poly(n)) but not exactly either. If there's a better term, that could help clarify the types of problems Theorem 2 targets because as of now, it is only including a (admittedly large) subset of commonly-seen DP problems.

4. For Theorem 2: is it possible to give a tighter bound than poly(M,N)? L817 suggests it is $NM^2$, is that a correct bound?


Minor comments that can be changed before final version:

* The results initially appear quite similar to Jalessi et al., 2024, and I still thought that even though the authors address this in the introduction, stating that Jalessi handles sequences does not represent the difference properly. I now believe the two papers and findings are different: Jalessi et al., 2024 focuses on scaling in terms of sequence length and number of states while this paper is focused distance of a token. Something in **bold** in the introduction could reduce this confusion.

* It would be nice to include an example of a DP problem in the appendix so that we don't need to open a separate paper to find one.

* Fix the BERT citation in L32

* There is some prior work/discussion around length (Ben-Kish et al., 2024 [link](https://arxiv.org/abs/2406.14528v1)) and "no free lunch" (Sec 5 of Mamba paper).

* Sec 4.1, when we set H_0=O, what is O? Do you mean 0?

* L212 Why is a_i \in [0,1]? I know why it is positive, but it isn’t clear to me why it’s <=1

* L718: le -> \le?

* L271: “given a input” -> “given an input”

* L279 “and” -> “and” (un-italicize it)

* L305: O(ploy(M, N) -> O(poly(m,n)) ["ploy(m,n)" appears multiple times elsewhere too]

* Sec 4.3 header: “Mamba **is** Empirically Weaker…”

* L347: “copy task finally, Transformer” is missing punctuation

* Update the theorem numbering in the appendix so they correspond with main body

---

> ### Author Response · Authors · 2024-11-20
>
> We greatly appreciate the reviewer's careful review and constructive suggestions. Below are our responses.
>
> > **W1:** While the inclusion of DP problems feels necessary to relate some of the abstract findings to a real problem (CoT with DP), the findings feel inconclusive because it teaches us about representational power and is an existence proof. However, as we observe in the Phonebook experiments, that doesn’t translate in practice to viability; in particular the bounds are still quite loose. When reading the paper, it felt like Section 5 was a separate paper as it introduces a new set of notation despite the result being ultimately a corollary of the original theorem.
>
> First, regarding these "abstract findings" about "representational power", we would interpret them as conclusions about the upper bound on the computational overhead required for the model to complete the task. Specifically, through constructive proofs, we provide an upper bound on the computational cost Mamba incurs when using CoT to solve DP tasks. Furthermore, concerning the distinction between "abstract findings" and "real problems," we believe adding more illustrative examples could help enhance the reader's understanding, as referenced in our response to **W2**. In fact, our initial intention was to follow the techniques of Feng et al. [1] to investigate the improvements in representational power brought by Mamba under CoT (Theorem 3 in ours and Theorem 4.7 in Feng et al. [1]). To explore Mamba block's ability to address DP problems, we required several lemmas (Theorem 2, Lemma 1-5) to demonstrate the fundamental operations Mamba blocks can perform (e.g., multiplication, approximate MLPs, linear transformations). During this process, we discovered that the COPY operation is critical for solving DP problems. Moreover, Mamba and Transformers exhibit significant differences in executing this operation, which led us to decide to dedicate a separate section to discussing Mamba's ability to perform the COPY operation.
>
> Therefore, the content in Section 4 is closely related to Section 5. Perhaps we should emphasize the differences between the two models in performing the COPY operation in Section 4, which could improve the overall organization of the content, as mentioned in our response to **Q1**.
>
> [1] Towards Revealing the Mystery behind  Chain of Thought: A Theoretical Perspective

---

> ### Author Response · Authors · 2024-11-20
>
> >  **W2：** It isn’t stated clearly why the assumptions (both Assumption 1 and Assumption 2) are safe to assume. When are these assumptions satisfied?
> >
> >  Assumption 1, first condition – I don’t understand why it is reasonable to assume this...
> >
> >  Assumption 1, second condition - this is missing a something. “There exists Si such that…” or “There exists j such that?
>
> For Assumption 1:  First, in the first condition, the introduction of $\rho$ and $\delta$ serves more as a convenient definition for the theorem's formulation (perhaps it should not have been presented as an assumption). We do not impose restrictions on the values of $\rho$ and $\delta$​, allowing them to be relatively flexible.
>
> For the second condition, as mentioned in Line 269, it is to ensure the stability of the output scale. This assumption is necessary; otherwise, during theoretical analysis, the output scale could grow significantly with the increase of $|S_i|$. In practice, this issue can be mitigated by using modules like layer normalization. However, such approaches would introduce additional complexity into the theoretical analysis. Hence, we adopted the current assumption.
>
> We believe there might be some misunderstanding regarding the issues mentioned in Assumption 1, which has led to us not fully grasping the point in the question. To clarify this, we would like to provide an example:  Given a text snippet, "XXXXXXX(Some unrelated content). Bob has a dog while John has a cat. After school, Bob often plays with his ____." We assume the expected output from the model here is "dog".
>
> In this example, assuming its position of the answer is $i$, the Matching Set $S_i$ for the current token to be predicted and $L \approx 18$ represents the distance within which the answer might exist relative to position $i$. In other words, starting from the phrase "Bob has a ...,". In addition, $pos(i)$ refers to the position index of the word "dog".  The Matching Set $S_i$ might include position indices for words like "Bob," "dog," "John," and "cat" if we choose some $\delta$. Of course, if we choose a larger $\delta$, the Matching Set $S_i$ might only include the position indices corresponding to "Bob" and "dog". Indeed, the suggestion that $\rho \ge \delta + \epsilon$ is a good idea for ensuring the "gap," and we will consider reorganizing and rephrasing this part in the future.
>
> For Assumption 2:  A significant number of DP problems conform to such conditions. For example, consider the problem of finding the minimum edit distance, where we aim to transform string $s_1$ into string $s_2$ with lengths $n_1 = |s_1|$ and $n_2 = |s_2|$, respectively. The costs for insertion, deletion, and substitution are $a$, $b$, and $c$, respectively. Let $\rm dp(j,k)$ represent the cost of transforming the first $j$ characters of $s_1$ into the first $k$ characters of $s_2$. The transition function can then be expressed as:
>
> $$
>  \text{dp}(j, k) =\begin{cases}    ak & \text{if} j = 0 \\\\    bj & \text{if } k = 0 \\\\    \min \big(         \text{dp}(j, k-1) + a, \text{dp}(j-1, k) + b,\\\\        \qquad \text{dp}(j-1, k-1) + c \mathbb{I}[s_j^{(1)} \neq s_k^{(2)}]    \big) & \text{otherwise}\end{cases}
> $$
>
> Finally, the aggregation function selects $\rm dp(n_1, n_2)$ as the final answer.  In the example above, the size of the state space, all intermediate values, and the lengths of the input strings will all be upper-bounded by $\rm poly(n_1, n_2)$. Moreover, the operations required by the involved functions can be approximated with polynomial efficiency by a constant-size MLP, as shown in Lemmas 1-5. Therefore, such DP problems satisfy Assumption 2.
>
> > **W3:** The presentation of the core theorems (Section 4.2) is very difficult to understand. It might be easier to present Theorem 1 in terms of δ (and use ρ only in the proof - the first condition doesn't actually need ρ either). For Theorem 2, the “simple intuition” could be better expressed in relation to the terms of Eq. 7, i.e. as a direct result of the first term increasing.
>
> Thank you for your suggestion. We have also noticed that the $\rho$ in Assumption 1 and Theorem 1 is redundant, and directly using $c_i^T b_{pos(i)}$ would indeed be more straightforward.  Regarding "as a direct result of the first term increasing," we do not understand your meaning as Theorem 1 deals with a constant-size setting relative to $N$ and the first term does not seem to reflect any explicit relationship with $N$.
>
> > **W4:** Finally, the empirical results would be significantly more interpretable if instead of the total number of parameters, the actual values of the Mamba block size were reported, since that's what the theoretical results are in reference to.
>
> We agree with your suggestion. Adding charts to illustrate the relationship between different block sizes (for example, 10M, 20M, ..., 130M, ...) and the number of training examples needed (just like Line 359) would better align with our theoretical results.

---

> ### Author Response · Authors · 2024-11-20
>
> > **Q1:** “free lunch” is not well-defined, ... Do you agree with this generalized statement? ... not misunderstanding the paper.
>
> What we intended to express with "Mamba always enjoys the free lunch" is that Mamba can accomplish a given task with a smaller model size (and inference time) compared to Transformers. To challenge this claim, in Section 4, we specifically focused on the defined COPY task, and showed the potential limitations of constant-sized models in reproducing historical information. In Theorem 1, we presented a sufficient condition for performing the COPY operation and pointed out that this condition is relatively difficult to satisfy. In such cases, **Mamba cannot enjoy the free lunch**.
>
> We believe the ambiguity in the "free lunch" statement might also stem from the lack of a direct comparison with Transformers in this section. In fact, whether Transformers can perform the COPY task is partly explained by Lemma C.7 of Feng et al., 2023 [1] which demonstrates that a constant-sized attention layer (relative to sequence length) can achieve COPY-like behavior, satisfying $\| o_i - v_{pos(i)} \| \le \epsilon$. However, for any given token, this operation incurs linear time overhead. Similarly, for Mamba to achieve the COPY task, it might also require a linear model size and corresponding linear inference time. From this perspective, **Mamba does not enjoy the free lunch** for this task. However, this comparison remains somewhat rough, as the formalized assumptions differ between the two models (e.g., Assumption 1 in our work for Mamba and Assumption C.6 in Feng et al. for attention layers), though the underlying logic bears similarities.
>
> In Section 5, we shifted focus to Mamba equipped with CoT, analyzing its upper bounds on time and space overhead when solving DP tasks compared to Transformers. This serves to show that **Mamba cannot always enjoy the free lunch**.
>
> We agree with your comment, *"If the distance ... recall that information,"* and believe your understanding is accurate.
>
> > **Q2:** What’s the significance of the $l_{\infty}$ norm in Theorem 2? Is it a corollary, assumption, or not actually relevant to the results?
>
> It can be understood as a corollary, where all parameters being upper bounded by $\rm O(poly(M, N))$ implies that the COPY task can also be solved by the Mamba block with $\rm log(N)$ precision (Line 310). For the reason behind using a $\rm log$-precision model, please refer to Appendix A.3 of Feng et al., 2023 [1]. Briefly, a model with $\rm log(N)$ precision can represent real numbers bounded by $\rm O(poly(n))$ with a truncation error of $\rm O(poly(1/n))$.
>
> > **Q3:** Is it possible to express the DP conditions of Assumption 2 in terms of a formal problem complexity class...including a (admittedly large) subset of commonly-seen DP problems.
>
> Assumption 2 follows the assumptions established by Feng et al., 2023 [1] and Yang et al., 2024 [2]. Under these same assumptions for DP problems, we can compare Mamba and Transformers. In the work of Merrill et al., 2023 [3], the capability upper bound of log-precision Transformers is $\rm TC^0$. Feng et al. demonstrated that Transformer+CoT can surpass this upper bound, but it still cannot solve the CFG problem (Theorem 4.8), which is $\rm P$-complete. Therefore, we intuitively believe that the problems described by Assumption 2 may lie between $\rm TC^0$ and $\rm P$-complete.
>
> [1] Towards Revealing the Mystery behind  Chain of Thought: A Theoretical Perspective
>
> [2] Do Efficient Transformers Really Save Computation?
>
> [3] The Parallelism Tradeoff: Limitations of Log-Precision Transformers
>
> > **Q4:** For Theorem 2: is it possible to give a tighter bound than poly(M,N)? L817 suggests it is NM2, is that a correct bound?
>
> As mentioned in our response to **Q2**, the parameter upper bound being $\rm poly(M, N)$ is intended to show that a $\rm log(N)$-precision Mamba block can perform this operation. While we are unsure if there exists a tighter bound, a $\rm log(N)$-precision Mamba will suffice in any case.  We are unclear about your question regarding "is that a correct bound." Are you suggesting that parameter sizes can be bounded by $\rm NM^2$, and that’s why we described it as $\rm poly(M, N)$? If so, that is exactly what we intended to convey.
>
> > **C5:** Sec 4.1, when we set H_0=O, what is O? Do you mean 0?
>
> Yes,  $\mathbf{O}$ means the matrix whose elements are all zeros and shape is same as $H_0$.
>
> > **C6:** L212 Why is a_i \in [0,1]? I know why it is positive, but it isn’t clear to me why it’s <=1
>
> Just as inllustrated in Line 208 & 209, all elements of $A$ are usually set to be negative while $\Delta>0$, thus $\mathrm{exp}(\Delta A) \le 1$.
>
> > **C1-C4 & C7-C13:** Advices and typos
>
> Thank you for your constructive suggestions and for carefully pointing out the typos. We will carefully adopt your recommendations and address these issues in subsequent versions.

---

> > ### Comment · Reviewer_k7wk · 2024-11-25
> >
> > Thank you for your detailed responses (and the responses to other reviewers) in answering my questions. I won't be changing my score due to W1 and W2. The title/intro promises a significant result but the assumptions that only hold in somewhat narrow cases. As the paper is currently written, it feels like it is proving a stronger result than prior work on copying, but I think the actual theorem is quite different a perhaps weaker (because of the conditions).
> >
> > Another possibility is framing the paper as a theoretical analysis of DP-like problems (or maybe analysis of $TC^0$ or $P$-complete problems). Starting from DP (as you explained in your response to W2), you could define the COPY task that is motivated by DP. This would be more independent of COPY literature in general and also tie in the DP theorems better. Finally then, there would be empirical results (additionally) on the DP problems.
> >
> > Finally, small comment on Q4: yes I think $O(NM^2)$ feels like a much tighter bound, than $poly(M,N)$, and in general it's preferred if the bounds were tighter if it doesn't require more effort to prove.

---

### Official Review · Reviewer_BtyZ · 2024-11-09

**Soundness:** 3
**Presentation:** 2
**Contribution:** 2
**Rating:** 3
**Confidence:** 4

**Summary:**

This paper explores the Mamba model's expressive power, particularly its performance on COPY tasks and dynamic programming (DP) problems. The authors show that, with linearly scaling size, Mamba can accurately execute COPY operations and, with Chain of Thought (CoT) prompting, can handle DP problems at a complexity similar to Transformers unless the problem has m-locality structure.

**Strengths:**

Many theoretical analyses are provided.

**Weaknesses:**

The theoretical results and empirical observations presented in this draft largely reiterate findings already covered in the literature, with minimal novel insights that would interest myself or other readers, leading me to recommend rejection.

Firstly, the theoretical contributions appear trivial. Although I am unsure why the authors define the COPY operation in its current form (see question 1), this task seems more akin to token-level (soft) associative recall. Stronger theoretical results on the state size requirements for RNNs (including Mamba) for recall tasks, such as Theorem 3.1 of Arora et al. (2024) [1] and Theorem 4.6 of Wen et al. (2024) [2], already establish that all RNNs require \(O(L)\) state size to address these tasks. Additionally, as acknowledged by the authors, Section 2.3 of Jelassi et al. (2024) [3] similarly concludes that achieving sentence-level COPY in Mamba (and other RNNs) requires linearly increasing hidden size. Given these precedents, I find the theoretical contribution of Theorems 1-2 here unclear in significance and unsurprising, as the proofs align closely with this existing body of work.

Secondly, regarding the CoT proof, Mamba can be formulated as a (gated) linear attention model, as detailed in Yang et al. (2024) [4]. This draft seems to restate established findings by connecting Mamba to linear attention and applying known results on Efficient Transformers (i.e., standard linear attention) from Wen et al. (2024) without adding unique insights. Thus, the paper’s contribution remains ambiguous.

Finally, this draft requires improvement in writing quality. The notation system is confusing; for example, it is unclear why in line 237, ¥$\alpha_j = \prod_{k=j+1}^i a_k$ is defined without any subscript or superscript in $\alpha$.

### Reference
[1] Simple linear attention language models balance the recall-throughput tradeoff https://arxiv.org/abs/2402.18668

[2] RNNs are not Transformers (Yet): The Key Bottleneck on In-context Retrieval  https://arxiv.org/abs/2402.18510

[3] Repeat After Me: Transformers are Better than State Space Models at Copying Transformers are Better than State Space Models at Copying https://arxiv.org/pdf/2402.01032

[4] Gated Linear Attention Transformers with Hardware-Efficient Training  https://arxiv.org/abs/2312.06635

**Questions:**

Regarding the COPY task setting, it appears that for each token $i$, a "relevant token" $\text{pos}(i)$ is randomly selected from a set $S_i$, defined such that the absolute value of $c_i^T b_{\text{pos}(i)}$ exceeds a threshold. Could you clarify the rationale behind this definition? I find it unclear why the COPY operation is structured this way and would appreciate an explanation of its intended purpose and relevance.

---

> ### Author Response · Authors · 2024-11-20
>
> We are grateful for the time and effort the reviewer have dedicated to our paper. Below are our responses.
>
> >**W1:** Firstly, the theoretical contributions appear trivial... Stronger theoretical results on the state size requirements for RNNs (including Mamba) for recall tasks...as the proofs align closely with this existing body of work.
>
> We believe there are significant differences here. Arora et al. [1] focus on a synthetic AR task called Multi-Query Associative Recall (MQAR), using communication complexity theory to show that recurrent models require at least $\Omega(N)$ to solve MQAR (Theorem 3.1). This provides a lower bound guarantee for solving the problem. In contrast, our Theorem 1 presents a sufficient condition for solving the COPY operation, and Theorem 2 provides an **upper bound** on the model size required to solve the COPY operation, meaning we use a constructive approach to show that the model size needed to complete the COPY task is at most $O(N)$.
>
> Similarly, in Wen et al.'s Theorem 4.6 [2], they express that even with CoT, any RNN model with $o(n)$ bit memory cannot solve tasks in $T \in \{\text{Index, AR, c-gram retrieval, Counting}\}$ of size $n$ for large enough $n$. They also use communication complexity and information-theoretic arguments to prove the lower bound. In comparison, our Theorem 2 does not involve the use of CoT for the Mamba block. Therefore, we believe the two conclusions are not in conflict.
>
> Additionally, Jalessi et al.[3] focus on scaling in terms of sequence length and the number of states, whereas our work focuses on analyzing the impact of the distance of the token to be copied on the output error from the **numerical approximation** perspective.
>
> Finally, regarding the statement "the proofs align closely with this existing body of work", we believe there has been a significant misunderstanding. As already explained, the works mentioned above mainly rely on communication complexity theory or information-theoretic results, while our approach is grounded in **numerical approximation**. We have not found direct connection between our proofs and the existing proofs in these works. Therefore, we believe our theoretical contributions are distinct and do not closely align with the existing bodies of works.
>
> [1] Simple linear attention language models balance the recall-throughput tradeoff
>
> [2] RNNs are not Transformers (Yet): The Key Bottleneck on In-context Retrieval
>
> [3] Repeat After Me: Transformers are Better than State Space Models at Copying
>
> > **W2:** Secondly, regarding the CoT proof, Mamba can be formulated as a (gated) linear attention model, as detailed ... the paper’s contribution remains ambiguous.
>
> Undoubtedly, there is a growing body of work linking Mamba with linear attention, such as the work by Yang et al. [1] mentioned, and those we referenced in Section 2 of our paper. In Section 3, we restate Han et al.'s work [2] simply to provide clarity and context for our readers, as also agreed upon by reviewer `PWf8`. This section is, of course, not our original contribution, as we have emphasized multiple times in the text, and it was not also presented as contribution in the introduction.
>
> Regarding the claim that we are "applying known results on Efficient Transformers (i.e., standard linear attention) from Wen et al. (2024) without adding unique insights," we believe this is a misunderstanding. In fact, in the CoT section, we follow the setup from Feng et al. [3]  focusing on Mamba's ability to solve the DP problem and comparing the results with those of standard Transformer [3] and efficient Transformers [4]. Our conclusions in this part are based on Theorem 2 and we did not rely on the conclusions from Wen et al. [5]. Our Theorem 3 provides an upper bound on the model size required for Mamba to solve the DP problem when equipped with CoT. This is distinct from Wen et al.'s Theorem 4.6, which provides a lower bound and is concerned with different problem settings. Therefore, we believe there has been a misunderstanding about our contributions and the used approach.
>
> [1] Gated Linear Attention Transformers with Hardware-Efficient Training
>
> [2] Demystify mamba in vision: A linear attention perspective.
>
> [3] Towards Revealing the Mystery behind  Chain of Thought: A Theoretical Perspective
>
> [4] Do Efficient Transformers Really Save Computation?
>
> [5] RNNs are not Transformers (Yet): The Key Bottleneck on In-context Retrieval

---

> ### Author Response · Authors · 2024-11-20
>
> > **W3:** Finally, this draft requires improvement in writing quality. The notation system is confusing; for example, it is unclear why in line 237, $\alpha_j= \Pi_{k=j+1}^i a_k$ is defined without any subscript or superscript in $\alpha$.
>
> Thank you for your suggestions. We will further refine our expression in the future to avoid causing confusion or misunderstandings for the readers. For the example you provided, $\alpha$ is simply a constant introduced for simplicity (using $\Pi_{k=j+1}^i a_k$ would seem redundant), and it carries no additional meaning.
>
> > **Q:** Regarding the COPY task setting, it appears that for each token i, a "relevant token" $\rm pos(i)$ is randomly selected from a set $S_i$, defined such that the absolute value of $c_i^T b_{pos(i)}$ exceeds a threshold. Could you clarify the rationale behind this definition? I find it unclear why the COPY operation is structured this way
>
> This definition is based on the attention perspective, meaning that when we want to recover historical records at some position $\rm pos(i)$ based on $c_i$ (query), the query ($c_i$) should maintain a sufficiently large attention score $c_i^T b_{pos(i)}$ with the key ($b_{pos(i)}$) corresponding to that position. This is also the difference in the starting point compared to the aforementioned work.

---

> > ### Comment · Reviewer_BtyZ · 2024-11-25
> >
> > Thank you for your explanation. However, I still cannot understand the motivation for designing such a task to distinguish between others (like whole sentence copy or associative recall). The space of trying to prove Mamba/Linear Attention/Linear RNN struggles from copy/recall/.. is rather crowded, and I foresee no significant insights provided by this paper. Regarding the second point, Wen et al 2024's construction for efficient linear transformer should be easily applicable to Mamba, a gated version of Linear Transformer (especially Mamba2 makes it more clear). Given this, the results aren't surprising at all. I am not a theory person, so I am unable to judge the theoretical significance of this work. However, as an efficient architecture practitioner, the results provided in this work give no insights and I thus recommend a clear rejection.

---

### Official Review · Reviewer_PqMr · 2024-11-09

**Soundness:** 2
**Presentation:** 2
**Contribution:** 2
**Rating:** 3
**Confidence:** 3

**Summary:**

This paper studies MAMBA's limitations in copying tasks. The authors start by introducing MAMBA basics and how MAMBA simulates linear attention. Then, the authors argue that, similar to linear attention, the dot-products between the current position and target positions should be larger than other positions in order to copy the target positions. Therefore, scaling MAMBA can improve copying performance as scaling is beneficial to linear attention.  Furthermore, the authors discuss how Chain-of-Thought can improve MAMBA in dynamic programming, but gains are limited by the length of the answer.

**Strengths:**

The topic is interesting. Models don't (currently) have an explicit blackboard to store variables for copying, only approximately through state vectors based on probabilistic predictions.

**Weaknesses:**

1. The presentation is not good. Notations are not concise enough for readers. One solution is, the authors can provide a notation list and figure out a way to share notations with literature. Figure 1 is a good idea. The authors could integrate equations into the figure.  I also feel some texts and equations are redundant in the main texts. For example, preliminaries (introduction to MAMBA) are too long and only Eq 1 is closely related to this paper. The paper should be more dense and concise.

2. Motivation is not clear. The authors claim that their work focuses on token-level tasks (line 115) in contrast to existing works. But all the discussions and experiments are based on copying a sequence. This can be seen in definition 1 "Then the output of COPY operation is a sequence of vectors o1,o2,...,oN with oi = vpos(i) where pos(i) ∈ Si is the position we want to copy".  Furthermore, it is not clear how copying connects to dynamic programming. Does the authors study in the setting that the model only do copy for DP?  Can you try some toke-level copying tasks like LAMBADA.

3. To me, the findings are not novel and not distinguished from existing works. Most of the paper focuses on introducing existing works.

4. The main claim about the bottleneck of MAMBA is not discussed in this paper.  While we can see this from empirical studies, it is not clear how we can take inspiration from the assumption the authors make.  This is important as the authors claim they explain the limitations from theoretical standpoints.

**Questions:**

Typos:
fI: In →In ? line 397
A(n) should be An. line 420.

I'm happy to discuss the paper in case of misunderstanding.

---

> ### Author Response · Authors · 2024-11-20
>
> We greatly appreciate the reviewer's careful review and constructive suggestions. Below are our responses.
> > **W1:** The presentation is not good. Notations are not concise enough for readers. One solution is, the authors can provide a notation list and figure out a way to share notations with literature. Figure 1 is a good idea. The authors could integrate equations into the figure. I also feel some texts and equations are redundant in the main texts. For example, preliminaries (introduction to MAMBA) are too long and only Eq 1 is closely related to this paper. The paper should be more dense and concise.
>
> Thank you for your suggestion. We will include a notation list in the subsequent version; we think it's a great idea. We have also noticed that some parts of Section 3 are indeed unnecessary, and we will streamline that section to avoid overloading the readers.
>
> > **W2:** Motivation is not clear. The authors claim that their work focuses on token-level tasks (line 115) in contrast to existing works. But all the discussions and experiments are based on copying a sequence...
> >
> > Furthermore, it is not clear how copying connects to dynamic programming. Does the authors study in the setting that the model only do copy for DP? Can you try some toke-level copying tasks like LAMBADA.
>
> In Section 4, the overall logic is as follows: for Mamba blocks of constant size, we have identified **sufficient conditions** for executing the COPY operation, as outlined in Theorem 1 and Line 297. We would like to emphasize that satisfying this sufficient condition is stringent. Of course, even in more relaxed situations where the sufficient conditions are not met, it is still possible for a Mamba block to perform the COPY operation. However, we want to point out that Mamba blocks of size $O(N)$ **always** can. While we need to provide a sequence $X=[x_i]_{i=1}^{N}$, the theoretical results focus on whether for each $x_i$, the historical record at $pos(i)$ can be copied, rather than copying the entire sequence as a whole.
>
> In Section 5, when solving the DP problem, the model can use the COPY operation to get answers for input or intermediate values to compute subsequent results. Our setup is not "only perform copy for DP," but rather the model solves various operations required in the DP process by combining the COPY operation with those demonstrated in the Lemmas in the appendix.
>
> As for LAMBADA, we directly tested the pretrained Mamba and Transformer models on the LAMBADA test data. For Mamba, we selected the three sizes used in the paper: 360M, 1.4B, and 2.8B. For the Transformer, we used the Pythia transformer models (Biderman et al., 2023) of sizes 410M, 1.4B, and 2.8B, as also adopted by Jalessi et al., 2024 [2]. We found that Mamba outperformed Transformer in terms of accuracy across all three sizes (for example, Mamba with 360M has 53% accuracy while Transformer with 410M has 44%). We suspect that this may be due to (1) we used the pretrained models directly without fine-tuning. Fine-tuning would involve appropriately segmenting the LAMBADA training text to create suitable training data (e.g., ensuring that the last word is meaningful and appears within a certain distance in the preceding text), and we have not yet performed training on LAMBADA's training text; (2) the distance between the target word to be predicted and its previous occurrence is not very large in the test data (the distance ranges from 10 to 80). We plan to further explore and design better approaches for token-level tasks in the future. For example, using tokens instead of a sequence of numbers in tasks like the Phonebook task might be a better choice.
>
> [1] Pythia: A suite for analyzing large language models across training and scaling.
>
> [2] Repeat After Me: Transformers are Better than State Space Models at Copying

---

> ### Author Response · Authors · 2024-11-20
>
> > **W3:** To me, the findings are not novel and not distinguished from existing works. Most of the paper focuses on introducing existing works.
>
> We have also noticed that due to insufficient explanation on our part, the conclusions of our work "seem somewhat similar" to those of other works, and the distinctions are not clearly highlighted. In fact, our work differs from existing works in terms of the tasks we focus on, the underlying intuition, and the theoretical results. Reviewer `vUfm` in Weakness 1, `BtyZ` in Weakness 1&2, and Reviewer `k7wk` in Minor Comment 1 also expressed similar concerns about the novelty of our work. We suspect that the "existing works" include the ones mentioned in their comments, and this part can refer to our corresponding responses. We will add a more detailed comparison with these works in the next version.
>
> > **W4:** The main claim about the bottleneck of MAMBA is not discussed in this paper. While we can see this from empirical studies, it is not clear how we can take inspiration from the assumption the authors make. This is important as the authors claim they explain the limitations from theoretical standpoints.
>
> As stated in our response to **W2**, the bottleneck logic we aim to express for Mamba is as follows: for Mamba blocks of constant size, we have identified **sufficient condition** for executing the COPY operation. We want to emphasize that satisfying the sufficient condition is stringent. Of course, even in more relaxed situations where the sufficient conditions are not met, Mamba blocks may still be able to perform the COPY operation. The question is, given any sequence and any position, is there always a constant-sized Mamba block that can execute the COPY operation? We have not provided an answer to this. However, we would like to point out that Mamba blocks of size $O(N)$ **always** can perform the operation. From this perspective, we state that constant-sized models **may** have a bottleneck in our paper. However, as pointed out by reviewer `PWf8`, we also recognize that this explanation is somewhat unclear and could be misleading for readers. We will revise this statement in the subsequent version to make it clearer. Furthermore, as with the theoretical analysis in previous work, our theory focuses on the expressive power of the Mamba block and whether this ability can be learned beyond the scope of this paper is a topic we will explore further in the future.

---

> > ### Comment · Reviewer_PqMr · 2024-11-25
> >
> > Thanks for your reply. I have no further quesionts.

---

### Official Review · Reviewer_PWf8 · 2024-11-10

**Soundness:** 2
**Presentation:** 2
**Contribution:** 3
**Rating:** 5
**Confidence:** 4

**Summary:**

This paper is mostly a theoretical study on the computational expressiveness of the Mamba model.
It studied Mamba's ability on a self-defined, non-standard "COPY" task and the computational cost needed to solve dynamic programming (DP) problems.
Under several important yet *opaque* assumptions, the paper draws several contingent conclusions ---

* Mamba *"may"* have trouble performing the "COPY" task reliably when the context to be copied can have arbitrary length (while the model is of a constant size)
* If the model size is allowed to scale linearly with context length to be copied, it can perform the "COPY" task perfectly
* Mamba requires a computational cost to Transformer to solve DP problems when equipped with CoT
* In DP problems with locality properties, Mamba can save computation

In addition to theoretical analysis, which comprises most of the paper, the authors also conduct empirical experiments that further sheds light on the realistic behaviors of Mamba on the COPY task.

**Strengths:**

* I appreciate the detailed yet concise background introduction to Mamba in Section 3 and 4.1.  These sections are well-written, set up good notation and summarize the literature on Mamba's architecture and its connection to linear attention well.

* Despite the existing theoretical work on SSM's ability to perform COPY task, the authors attempted to formulate a different kind of COPY task and studies Mamba's ability to solve it.  This formulation goes beyond the discretized formulation of the COPY in prior work and study the numerical approximation aspect (however, to make this possible, the authors introduces several *significant* caveats that I'll detail in Weakness section).

* Studying the computational complexity of Mamba using the connection between SSM and linear attention is novel.  This angle allows the authors to adapt the proofs in Feng et al. (2024) that were intended for the softmax-attention in the standard Transformer.

* To the extent of the portions that I have checked, the mathematical derivations seem correct (this does not mean that the results are significant).

**Weaknesses:**

My main issue is that the theoretical results presented in the paper not only comes with significant caveats (in the form of assumptions) that are very hard to interpret, but also have conclusions that are hard to interpret and may be too weak to draw conclusions from.  These issues are somewhat swept under the rug by the authors since they did not discuss the implications.

Starting with Definition 1, it appears that the set $S_i$ is actually dependent on a specific instance of Mamba model since the definition of the COPY task involves the intermediate hidden states $c_i$ and $b_i$.  Furthermore, the COPY task is limited to the cases where the positions to copy must be in the set that's defined *with respect to* a specific instance of Mamba model.  The fact the task is dependent on the specific model parameters seems circular and hard to interpret, but the authors made no effort in discussing or disclosing this caveat.

Aside from the previous hidden assumption, the actual Assumption 1 is also opaque.  It's very unclear when an actual Mamba instance will satisfy this assumption, and without such discussion, the theorems derived under this assumption should not be used to draw even more general conclusions, such as "Mamba models may encounter bottlenecks when handling COPY task".  The assumption itself is also written in rather confusing language.  I spent a long time trying to figure out the dependence of the variables.

Finally, theorem 1 states that there is a specific Mamba model that can perform COPY but its lower bound $\rho$ can blow up.  What it *doesn't* say, is that any Mamba model that can perform COPY (up to $\epsilon$) will suffer from this exponential blow up.  The statement of the theorem seems too weak that the later "conclusion" seems misleading.  I interpreted this as -- Given the task and $L$, one can find a Mamba block that approximates the COPY task well; but the theorem says nothing about the rest of the Mamba blocks in the hypothesis space.  This is in contrast to previous work, such as Jelassi et al. (2024), who provided universal error bounds for SSMs on the COPY task.

Overall, the dependence between variables should be made clear (See concrete confusion points in the Questions section).
The authors can perhaps think deeper about the effects of the caveats and incorporate such thinking into the writing.
Moreover, the authors should more explicitly frame the limitations of their conclusions, so that readers won't think their time is wasted after they decided to dig deeper into paper based on the abstract or introduction.  I'd be glad to update my scores if these concerns are addressed.

**Questions:**

As written, variables in definitions/theorems/assumptions are so entangled that, as a reader, it is quite hard to understand their logical relationships, e.g.

* the task definition depends on $\delta$, but then the theorem states that $\delta$ observes some constraint; it would be helpful to discuss this circular dependence

* the task definition depends on $b_i, c_i$ which depends on the Mamba block, but then the theorem shows the existence of a Mamba block given the task.  Again, it would be helpful to discuss this circular dependence

Another concrete way to improve these confusion points is to stay close to standard mathematical language, such as "Given ___, there exists ___, s.t. ___ holds" and ensure that the assumptions are actually made clear.  For example, Definition 1 probably should include "Given a Mamba block and ..."; Assumption 1 first part can be written as "There exists $\rho$ such that for any $i \in [N]$, ... holds".

---

> ### Author Response · Authors · 2024-11-20
>
> We sincerely thank the reviewer for the constructive questions and suggestions. We have noted the reviewer's confusion regarding the circular dependence of definition mentioned in the weakness section. We would like to firstly provide a more detailed discussion of **Q1** and **Q2**.
> > **Q1:** the task definition depends on $\delta$, but then the theorem states that $\delta$ observes some constraint; it would be helpful to discuss this circular dependence
>
> You may understand it as follows: Under the premise of Assumption 1, when a Mamba block **additionally** satisfies Eq. (7), it can perform the COPY operation. In this case, under the constraints of Assumption 1, it naturally follows that $\delta \leq \frac{\epsilon}{(L-1)M||\Delta||\_{\infty}}$​, as demonstrated in the proof.
> In Assumption 1, we did not impose specific restrictions on the values of $\rho$ and $\delta$, only the relative relationship $\rho \geq \delta$. Under the scenario described in Theorem 1, it further has that "$\rho \geq \text{the right-hand side of Eq. (7)} \geq \delta$" **and** "the second condition of Assumption 1" $\Rightarrow$ "$\delta \leq \frac{\epsilon}{(L-1)M|| \Delta|| \_{\infty}}$".
>
> > **Q2:** the task definition depends on bi,ci which depends on the Mamba block, but then the theorem shows the existence of a Mamba block given the task. Again, it would be helpful to discuss this circular dependence
>
> The definition of the COPY operation indeed depends on different Mamba blocks: the parameters of different Mamba blocks will yield different attention scores $c_i^T b_j$ for all $i \le N$ and $j \le i$, as well as different $v_i$ for all $i$. However, as long as their output satisfies $o_i \approx v_i$, we consider that they have performed the COPY operation. When the input sequence $X = [x_i]_{i=1}^{N}$ and the position $pos(i)$ we aim to copy is fixed, additionally given some $\delta$ and we consider those Mamba blocks that satisfy Assumption 1, what are the **sufficient conditions** for these Mamba blocks to achieve the COPY operation? We aim to identify a subspace of the hypothesis space that satisfies these conditions, and this discovery is existential. We suspect that much of the confusion may stem from the role of $\delta$. As explained in the response to **Q1**, if the initial value of $\delta > \frac{\epsilon}{(L-1)M||\Delta||\_{\infty}}$, then either Assumption 1 or Eq. (7) will not be satisfied.
>
> ---
>
> In addition, we have also noted the reviewer's concerns about the limitations of the theoretical results mentioned in the weakness section. Below is our response.
> > **Weakness part**：More disccsuion on Limitations of Theorem 1 & Better writing presentation
>
> For Mamba blocks of constant size, we have identified the **sufficient conditions** for performing the COPY operation, as stated in Theorem 1 and Line 297. We would like to point out that satisfying these sufficient conditions is stringent because we are considering a relatively general scenario for a given sequence $X = [x_i]_{i=1}^{N}$ and $pos(i)$. Of course, even under less restrictive conditions that do not meet the sufficient criteria, it is still possible for a Mamba block to achieve the COPY operation.
> However, the underlying question is: for any given sequence and position, is it always possible for a constant-sized Mamba block to perform the COPY operation? We have not provided a definitive answer. However, we want to emphasize that Mamba blocks of size $O(N)$ **can always** achieve this.
>
> Our expression is indeed somewhat unclear. In the next version of the paper, we will revise the relevant sections based on your valuable feedback and that of other reviewers. We will reorganize the language used in the definitions, assumptions, and theorems to ensure clearer and more rigorous expression. Additionally, we will provide a more thorough discussion of the limitations of the theoretical results to avoid confusion or misunderstanding for the readers.

---

> ### Comment · Reviewer_PWf8 · 2024-12-03
> **Thank you for the clarification**
>
> I appreciate the clarifications from the authors and the effort they have invested in this work.  I think the paper cannot be published as-is (because, judging from confusion it has generated among the reviewers including myself, this work will lead to a lot of reader confusion).  But, I believe that, with major revision and improvements on presentation, clarity and framing, this work will become valuable for researchers in relevant domains.  I look forward to the next version of this work.

---

### Meta-Review · Area_Chair_56fX · 2024-12-22

**Metareview:**

### Summary:
This paper investigates the expressive power of the Mamba model, focusing on COPY operations and dynamic programming (DP) problems. Main finding include: with linearly scaling size, Mamba can accurately execute COPY operations and, with Chain of Thought (CoT) prompting, can handle DP problems at a complexity similar to Transformers unless the problem has m-locality structure. The authors provide both theoretical analyses and empirical results on synthetic COPY tasks to support their claims.

### Strengths:
This paper presents many theoretical analyses of mamba.

### Weaknesses:
The theoretical analysis is based on many complex assumptions. The theoretical findings are incremental to existing works.

**Additional Comments On Reviewer Discussion:**

1. Ambiguity of assumptions and theoretical results: reviewers generally find that the assumptions laid out are difficult to justify.
2. Overlap with existing literature: this is the biggest point raised during rebuttal. Reviewers noted that the findings in this paper don't bring new insights and largely overlap with existing findings.
3. Lack of empirical results for the DP problem.

In general, authors tried to clarify these points but failed to fully address these concerns.

---

### Decision · Program_Chairs · 2025-01-22

Reject